# Robust and tunable signal processing in mammalian cells via engineered covalent modification cycles

Ross D. Jones [1,2], Yili Qian [2,3], Katherine Ilia[1,2], Benjamin Wang[2,4], Michael T. Laub [2,4,5], Domitilla Del Vecchio [2,3 ✉] & Ron Weiss [1,2,6 ✉]

Engineered signaling networks can impart cells with new functionalities useful for directing differentiation and actuating cellular therapies. For such applications, the engineered networks must be tunable, precisely regulate target gene expression, and be robust to perturbations within the complex context of mammalian cells. Here, we use bacterial two-component signaling proteins to develop synthetic phosphoregulation devices that exhibit these properties in mammalian cells. First, we engineer a synthetic covalent modification cycle based on kinase and phosphatase proteins derived from the bifunctional histidine kinase EnvZ, enabling analog tuning of gene expression via its response regulator OmpR. By regulating phosphatase expression with endogenous miRNAs, we demonstrate cell-type specific signaling responses and a new strategy for accurate cell type classification. Finally, we implement a tunable negative feedback controller via a small molecule-stabilized phosphatase, reducing output expression variance and mitigating the context-dependent effects of off-target regulation and resource competition. Our work lays the foundation for establishing tunable, precise, and robust control over cell behavior with synthetic signaling networks.

[1] Department of Biological Engineering, Massachusetts Institute of Technology, Cambridge, MA 02139, USA. [2] Synthetic Biology Center, Massachusetts Institute of Technology, Cambridge, MA 02139, USA. [3] Department of Mechanical Engineering, Massachusetts Institute of Technology, Cambridge, MA 02139, USA. [4] Department of Biology, Massachusetts Institute of Technology, Cambridge, MA 02139, USA. [5] Howard Hughes Medical Institute, Massachusetts Institute of Technology, Cambridge, MA 02139, USA. [6] Electrical Engineering and Computer Science Department, Massachusetts Institute of Technology, Cambridge, MA 02139, USA. ✉email: ddv@mit.edu; rweiss@mit.edu

Across all organisms, sensing and processing of environmental factors are critical for growth, proliferation, and survival[1]. Engineering mammalian cells to transmute specific intracellular and extracellular inputs into desirable output behaviors has broad applications in cell therapy, biomanufacturing, and engineering stem cells, tissues, and organoids[2–8]. Recently, work has accelerated to rewire natural signaling pathways and engineer synthetic receptors that sense extracellular inputs[9–11]. A desirable engineered signaling system would have tunable input/output responses, low output noise, and drive gene expression that is robust to perturbations coming from the extracellular, cellular, and genetic context of the system[12]. The ability of the signaling system to exhibit such properties depends on how input signals are processed to generate gene expression outputs. However, relatively little work has been done to engineer such signal processing behavior in mammalian cells.

To date, nearly all engineered signaling systems have utilized either native intracellular signaling domains or proteolytic mechanisms to transduce extracellular signals into intracellular responses[11]. Interfacing with the cell's natural signaling networks has been a powerful method to rewire signaling pathways[10], but it is difficult to modulate signaling between natural receptors and their gene expression targets due to the complexity of natural signaling networks in mammalian cells. Using proteolysis to liberate gene regulators from the plasma membrane enables regulation independent from the cell signaling context through non-native proteins such as dCas9 or tTA[10,11]. However, because the effector proteins are irreversibly released from the receptor and thus the signaling system cannot be easily reset to its initial state, the ability to tune the input-output response is limited. Recently, an alternate design strategy based on non-native protein phosphorylation has been realized by fusing extracellular receptors or dimerization domains to bacterial two-component signaling (TCS) proteins; such receptors were shown to successfully transmute extracellular ligand inputs to synthetic TCS-regulated transcriptional outputs in mammalian cells[13,14].

The use of TCS proteins in synthetic mammalian signaling networks has the potential for creating tunable, robust signaling circuits that do not cross-react with existing networks in mammalian cells. Although TCS pathways are ubiquitous in bacteria, with dozens in *Escherichia coli* alone, they are generally rare in eukaryotes and absent in animals[15]. TCS pathways typically comprise a transmembrane sensor protein called a histidine kinase (HK) and a cognate intracellular effector protein called a response regulator (RR). In response to specific signal inputs, the HK autophosphorylates on a conserved histidine residue and then transfers the phosphoryl group to a conserved aspartate residue in the receiver (Rec) domain of the RR (referred to as the HK's kinase activity). Once phosphorylated, most RRs carry out transcriptional regulation, though other modes of regulation are also possible[16,17]. Unlike typical eukaryotic receptors, in the absence of signal inputs, most HKs directly catalyze the removal of the phosphoryl group from their cognate RRs (referred to as the HK's phosphatase activity)[17,18]. The presence of signal input alters the conformational state of the HK, thereby tuning its relative kinase and phosphatase activities[19]. The bifunctional nature of HKs is important for insulating TCS pathways from off-target interactions[20,21] and increasing the responsiveness to signal inputs[22]. The recently developed TCS-based receptors work by coupling ligand-induced dimerization of the receptor to HK kinase activity and thus RR-driven gene expression[13,14]. The lack of any known examples of histidine-aspartate phosphorelays in mammalian cells strongly suggests that these introduced TCS signaling pathways are insulated from mammalian signaling pathways[13,14,23].

Here, we introduce a framework for engineering signal processing circuits in mammalian cells based on synthetic covalent modification cycles (CMCs) built with bacterial TCS proteins (Fig. 1). In phosphorylation cascades, phosphatases that are constitutively active or part of a negative feedback loop can impart tunability and robustness to perturbations into the system through the reversal of substrate phosphorylation[24–28]. To develop such circuits, we isolate monofunctional kinases and phosphatases from the bifunctional *E. coli* HK EnvZ[29], then use specific phosphorylation and dephosphorylation of EnvZ's cognate RR OmpR to regulate transcriptional activation of downstream gene expression outputs. First, we illustrate the tunability of this system by using the level of an EnvZ phosphatase to shift the sensitivity of OmpR-driven gene expression output to the levels of an EnvZ kinase. Further, we show that kinase-to-output dose–responses can be tuned by regulating phosphatase expression with small molecule-inducible degradation domains. We then build upon this tunability to create phosphorylation-based miRNA sensors that are capable of cell type classification and enable cell-type-specific tuning of signaling responses.

In addition to making tunable sensors, a major general challenge for developing synthetic genetic circuits is undesirable context-dependence due to factors such as off-target binding of gene regulators and overloading of cellular factors used in gene expression (i.e. resources), which can perturb gene expression levels[30,31]. At present, there is a lack of synthetic signaling circuits that are robust to such context effects in mammalian cells. To address this problem, we used negative feedback control to impart robustness to perturbations into the kinase-to-output process of our circuit. The negative feedback is achieved by co-expressing the output protein with a phosphatase that dephosphorylates OmpR, returning it to an inactive form. The feedback strength and output level can be tuned via a small molecule-inducible degradation domain fused to the phosphatase. The addition of feedback control substantially reduces cell-to-cell noise in output expression and mitigates the effects of off-target post-transcriptional repression and loading of transcriptional resources on the signaling input-output response. Overall, we present the design and characterization of phosphorylation-regulated genetic modules that enable tunable, precise, and robust control of signaling outputs in mammalian cells.

## Results

**Engineering EnvZ to isolate kinase and phosphatase activity.**
As a model system for engineering synthetic signal processing circuits, we utilized the well-characterized EnvZ-OmpR TCS pathway from *E. coli*[32]. Like many HKs, EnvZ is bifunctional, actuating both kinase and phosphatase activity onto its cognate RR OmpR[29]. We thus reasoned that we could isolate the individual kinase and phosphatase activities of EnvZ to generate enzymes suitable for implementing a CMC. In particular, we expected that by starting with a bifunctional enzyme, we could selectively mutate or otherwise disrupt the kinase or phosphatase activity of EnvZ, yielding an enzyme significantly biased towards one activity or the other (Fig. 2a). Both in vitro and in vivo in bacteria, it has been shown that this objective can be achieved through various mutations[33–35], truncations[36,37], and domain rearrangements[38]. In mammalian cells, it was shown that wild-type (WT) EnvZ is constitutively active[23], indicating that it has net-kinase activity. However, we hypothesized that EnvZ may still retain some phosphatase activity and thus not operate as potently as a pure kinase. To begin creating more monofunctional kinases and phosphatases from EnvZ in mammalian cells, we generated several variants of EnvZ using established mutations, truncations, domain rearrangements, and novel combinations thereof (Fig. 2b–d).

To test for improved kinase activity, we evaluated the ability of EnvZ variants to activate an OmpR-driven reporter when

## a Engineered signal processing via covalent modification cycles

**Fig. 1 Overview of engineered covalent modification cycle. a** A covalent modification cycle (CMC) is composed of a substrate that is interconverted between an active and an inactive form by two different enzymes. Here we examine a CMC created by reversible phosphorylation/dephosphorylation of a transcription factor (TF) by a kinase and phosphatase. The inputs to this CMC, $u_K$ and $u_P$, alter the production rate or catalytic rates of the kinase and phosphatase, respectively. The output(s) of the system are RNA and/or protein species, whose production is activated by the TF when phosphorylated. Closed-loop (CL) negative feedback control can be achieved by co-expressing a phosphatase with the output. Without the feedback, the expression of the outputs is open-loop (OL). **b** The input/output (i/o) response of the system, i.e., the response of the TF-driven output(s) to kinase inputs ($u_K$), can be tuned via phosphatase inputs ($u_P$). Brighter lines correspond to the increasing concentration or activity of the phosphatase. **c** Negative feedback is expected to convert multimodal output responses into unimodal responses. Here, brighter lines correspond to increasing kinase concentration/activity. **d** Negative feedback is expected to impart robustness to perturbations in the output production process. The setpoint refers to the level of output in the absence of a perturbation.

transfected into HEK-293FT cells (Fig. 2e, left). OmpR-activated promoters were made by placing three to nine OmpR binding sites upstream of a minimal CMV promoter or a synthetic minimal promoter (YB_TATA[23], referred to as minKB), of which the 6xOmpR_BS-minCMV variant was chosen for use in most downstream experiments due to its high fold-change in response to OmpR phosphorylation (Supplementary Figure 1). From this initial screen, we identified two EnvZ variants, EnvZm2 [T247A] and EnvZm2[AAB], the latter having an extra DHp domain fused to EnvZ[223+][38], that induced higher levels of output expression than WT EnvZ, suggesting that their phosphatase activity is reduced (Fig. 2e, right; see Supplementary Figure 2 for further experimental details and data analysis). Variants expected to be deficient in ATP binding or autophosphorylation based on previous studies in bacteria were indeed found to lack activation of OmpR-VP64, indicating that in mammalian cells they also lack kinase activity (Fig. 2e, right). Moving forward, we used EnvZm2 as our kinase of choice due to its improvement in kinase activity and the highly conserved ability of the T247A mutation to reduce or eliminate phosphatase activity in other HKs[39,40].

To test for phosphatase activity, we co-expressed EnvZm2 with OmpR-VP64 to generate phosphorylated OmpR-VP64 (P-OmpR-VP64), and then evaluated the ability of our EnvZ variants to deactivate the expression of an OmpR-driven reporter (Fig. 2f, left). Although several EnvZ variants predicted to be phosphatases based on previous studies indeed showed deactivation of OmpR-driven expression at high concentrations, this deactivation was comparable to that of a variant mutated to

eliminate all catalytic activity (EnvZm0m1m2m3 [H243A/ D244A/T247A/N343K]) (Fig. 2f, right; & Supplementary Figure 3). Thus, it is possible that these variants were primarily inhibiting output expression through sequestration of P-OmpR-VP64 from its target promoter, rather than through dephosphorylation. Indeed, we found that high dosages of one such variant (EnvZm1, [T247A]) reduces "leaky" activation of the output reporter by non-phosphorylated OmpR-VP64, indicating that the observed reduction in output can occur absent dephosphorylation (Supplementary Figure 4). Notably, at both low (Fig. 2f) and high (Supplementary Figure 4) dosages of the variant EnvZ[A] (DHp domain only), we found no apparent phosphatase activity, contrasting with the original report[37]. Only variant EnvZm1[AAB], having an extra DHp domain fused to EnvZ[223+] with the mutation [D244A] in both DHp domains, was found to deactivate OmpR-driven expression more strongly than EnvZm0m1m2m3 (which lacks catalytic activity) (Fig. 2f, far-right), suggesting EnvZm1[AAB] has phosphatase activity in mammalian cells. However, at higher dosages of EnvZm1[AAB] and in the absence of EnvZm2, OmpR-VP64 appears to become activated, indicating that this variant may still retain some kinase activity (Supplementary Figure 4). We thus sought another means to generate a strong monofunctional EnvZ phosphatase that is functional in mammalian cells.

**Derivation of strong EnvZ phosphatases through DHp domain rotations.** Because of the constitutive kinase activity of WT EnvZ

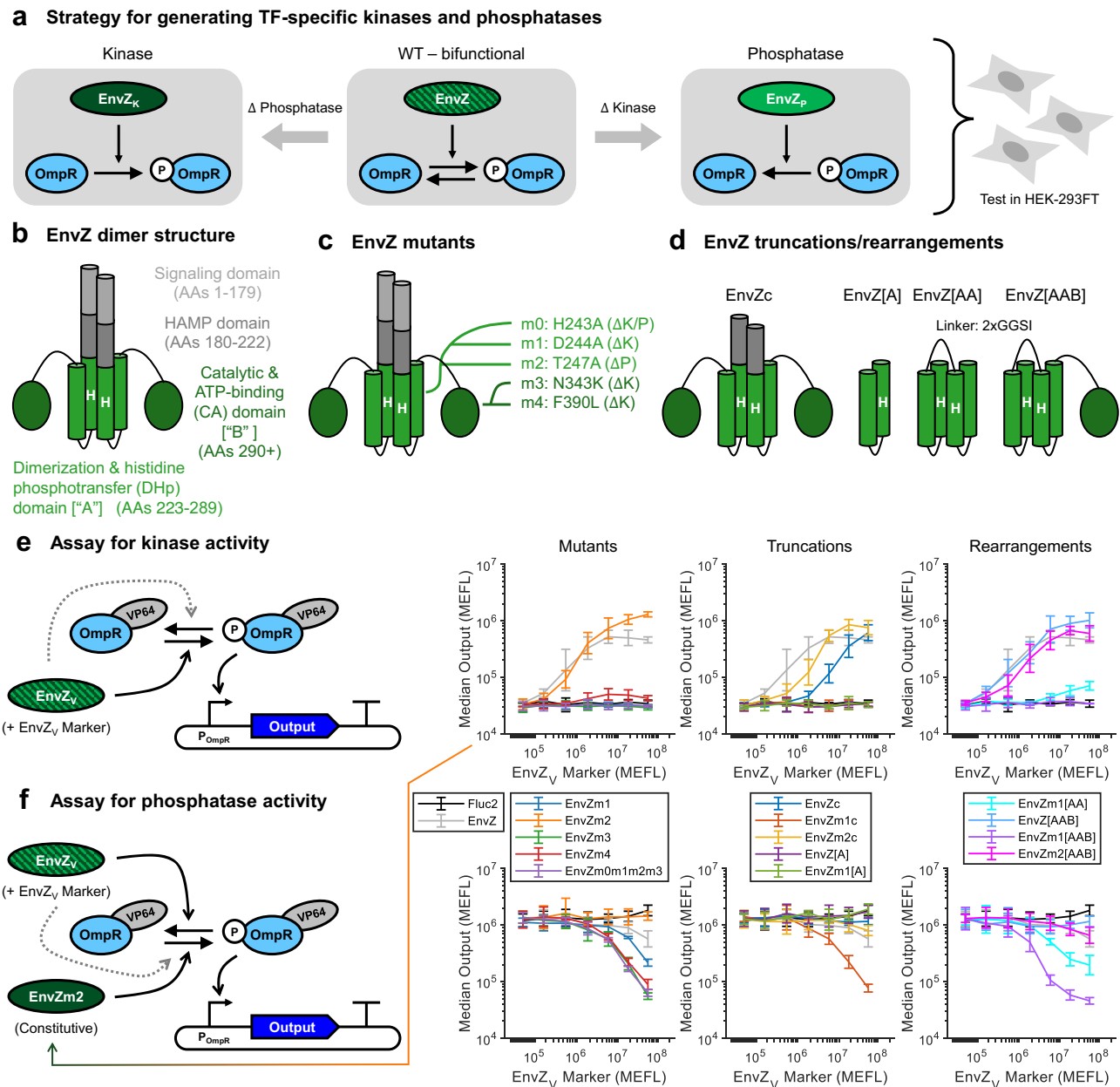

**Fig. 2 Engineering the bifunctional histidine kinase EnvZ to isolate kinase and phosphatase activities. a** Model system to construct a covalent modification cycle (CMC): EnvZ/OmpR proteins from *E. coli* two-component signaling (TCS). EnvZ naturally exhibits both kinase and phosphatase activities. By mutating or changing the enzyme structure, either one or the other activity is eliminated to yield a kinase or a phosphatase specific for OmpR. **b** Schematic of EnvZ. In *E. coli*, the signaling domain senses increasing environmental osmolarity, which is transmuted via the HAMP domain to the rest of the structure. Upon receiving this input signal, the CA domain ["B"] binds to ATP and autophosphorylates a conserved histidine residue in the DHp domain ["A"]. This phosphate group is then transferred to the cognate response regulator, OmpR. In the absence of signal input, the DHp domain catalyzes dephosphorylation of OmpR. In mammalian cells, wild-type EnvZ is constitutively active[23]. **c** Mutants of EnvZ[33–35,40,101] tested in this study, mapped onto their location in the EnvZ structure. **d** Truncations[36,37] and domain rearrangements[38] of EnvZ tested in this study. **e** Evaluation of EnvZ variants for improved kinase activity. Constitutively expressed OmpR-VP64 (OmpR fused to the activation domain VP64) was co-transfected with a reporter plasmid comprising a promoter ($P_{OmpR}$) with 6xOmpR binding sites and a minimal CMV promoter driving TagBFP as the output. Each EnvZ variant ($EnvZ_V$) was co-delivered with $EnvZ_V$ Marker, a fluorescent reporter that indicates dosage per cell, and poly-transfected against the other plasmids to evaluate the $EnvZ_V$-to-output dose–responses (see Supplementary Figure 2 for additional details). The solid arrow from $EnvZ_V$ to the phosphorylation cycle indicates the desired kinase activity, while the dashed arrow indicates the possibility of residual phosphatase activity. The plots on the right show the expression of OmpR-driven TagBFP (Output) in response to increasing $EnvZ_V$ dosages, as measured by $EnvZ_V$ Marker. **f** Evaluation of EnvZ variants for phosphatase activity. The experiment is similar to **e**, except that the variant EnvZm2, which has a strong kinase bias, is constitutively expressed to set a baseline level of OmpR phosphorylation. The solid and dashed arrows are swapped to indicate the desire for phosphatase activity and the possibility of residual kinase activity. All data in **e** and **f** were measured by flow cytometry at 48 hours post transfection in HEK-293FT cells. All error bars represent the mean ± s.d. of measurements from three experimental repeats. Source data are provided as a Source Data file.

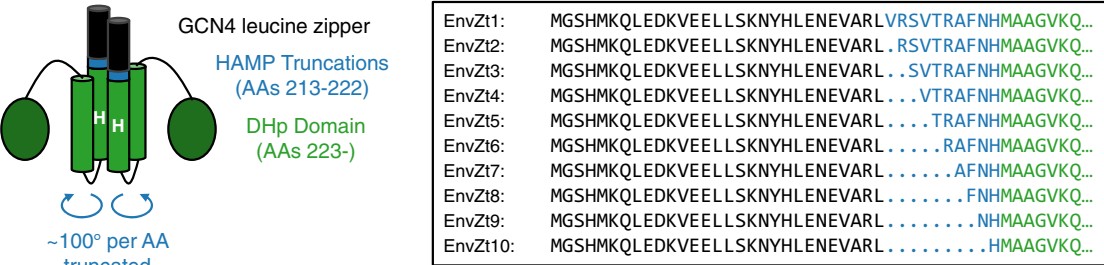

**a  Tuning enzyme activity through DHp domain rotations**

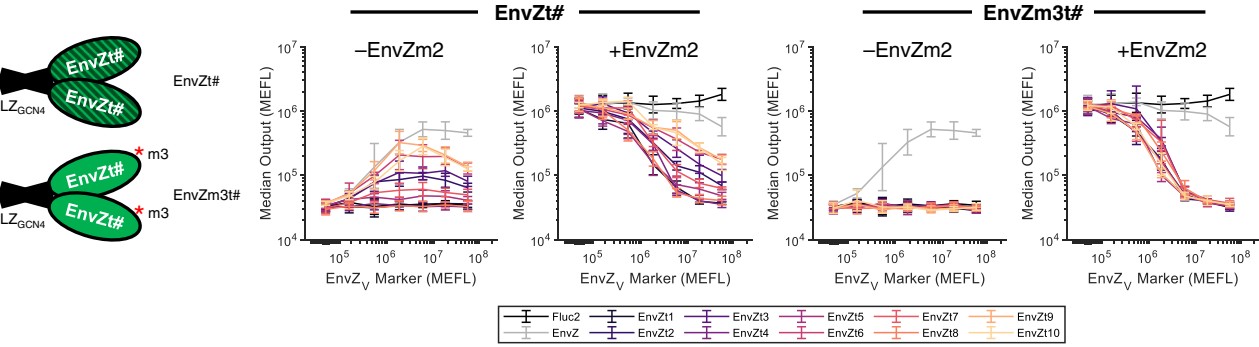

**b  Kinase and phosphatase activity assays**

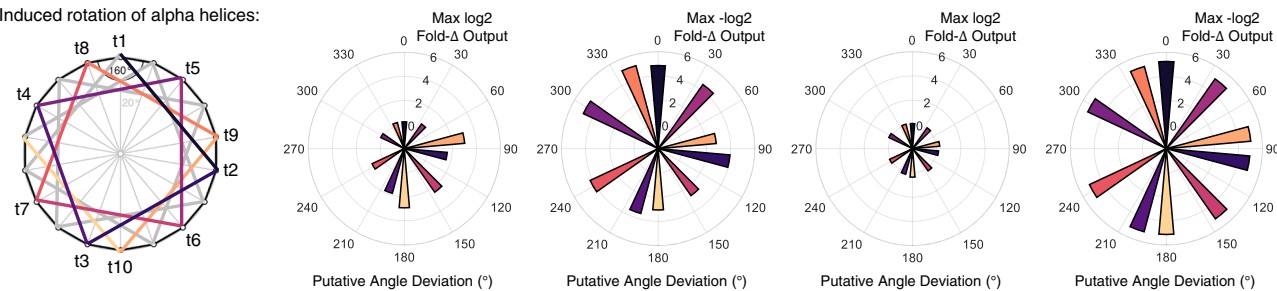

**c  Angular mapping of enzymatic activities**

**Fig. 3 Fixing DHp domain rotation to tune kinase activity and isolate strong phosphatases. a** Fusion of GCN4 leucine zipper to the N-terminus of EnvZ truncated between residues 212 and 221, thus connecting to the DHp domain and fixing the rotation of its alpha helices[41]. The box on the right shows the sequence of the first 37–46 amino acids of each variant. **b** Kinase and phosphatase activity assays for each rotationally locked variant (same experimental design as in Fig. 2e–f). EnvZm2, a kinase-biased variant, establishes a baseline of phosphorylated OmpR for testing dephosphorylation. EnvZm3t# have mutation 'm3' (N343K), which knocks out kinase activity. **c** Maximum fold-change in output expression induced by each variant mapped to the putative rotational conformation of the DHp domain, assuming 100° of rotation for each amino acid truncated between GCN4 and the DHp domain and setting EnvZt1 to 0°. All data were measured by flow cytometry at 48 hours post transfection in HEK-293FT cells. All error bars represent the mean ± s.d. of measurements from three experimental repeats. Source data are provided as a Source Data file.

and the lack of clear monofunctional phosphatase activity by purported phosphatase variants of EnvZ, we hypothesized that in mammalian cells, EnvZ may take on a structural conformation that is unfavorable for phosphatase activity. Previously, it was shown that the autophosphorylation rate of another HK, AgrC, can be modulated by changing the rotational state of the alpha-helices comprising the DHp domain[41]. We hypothesized that this rotational conformation may likewise affect access to the phosphatase state. We therefore followed the approach of Wang et al.[41] to force the alpha-helices in the DHp domain of EnvZ into fixed rotational states using GCN4 leucine zippers (Fig. 3a).

We generated a library of 10 rotationally-locked variants (EnvZt#1–10) with and without a mutation to eliminate ATP binding and hence kinase activity (m3 – [N343A])[34]. As expected, we observed a range of OmpR-driven gene expression levels that depend on the putative rotational angle of the DHp domain

(Fig. 3b and c). Interestingly, compared with WT EnvZ, all of the EnvZt# variants drive equivalent or weaker output activation by OmpR-VP64, while also reducing EnvZm2-induced expression by at least threefold (Fig. 3b). Comparing the exact levels of output with and without EnvZm2, we found that EnvZt# variants are capable of overriding the initial phosphorylation state of OmpR-VP64 to ultimately set a defined level of output (Supplementary Figure 5). Most strikingly, all EnvZm3t# variants showed potent and nearly identical deactivation of OmpR-driven expression back to baseline levels, regardless of their rotational conformation (Fig. 2c), indicating that all GCN4-fused truncations possess similar phosphatase activities. These data suggest that the fusion protein itself takes on a conformation that is amenable to phosphatase activity, possibly due to the formation of a more rigid structure[19], whereas the rotational state of the DHp domains primarily affects autophosphorylation.

To more quantitatively compare the activation and deactivation of OmpR-driven expression by each of the EnvZ variants described above, we fit simple first-order models to estimate the dosages of each variant needed for half-maximal activation or deactivation ($K_{1/2}$) of the output (Supplementary Figure 6). Notably, the EnvZm3t# variants deactivated output expression with $K_{1/2}$ values two- to threefold smaller than our previous best putative phosphatase, EnvZm1[AAB], and 10- to 20-fold smaller than the enzymatically null variant EnvZm0m1m2m3 (Supplementary Figure 6), indicating potent phosphatase activity. Moving forward, we chose to use the variant EnvZm3t10 as our phosphatase because it has one of the lowest values of $K_{1/2}$ among all EnvZ variants and completely deactivates the output down to basal levels (Fig. 3b and Supplementary Figure 6).

To ensure that the observed putative phosphatase activity is not explained by the formation of partially or completely inactive heterodimers between any putative phosphatases and EnvZm2, we repeated the experiments described above with CpxA in place of EnvZm2 (Supplementary Figure 7a). CpxA has weak off-target kinase activity for OmpR[21] and, broadly, heterodimerization between different HKs is rare[42]. In the presence of CpxA, the putative phosphatases similarly, and in some cases more potently, deactivate OmpR-driven expression (Supplementary Figure 7b, c). Thus, the observed output deactivation is independent of how OmpR-VP64 is phosphorylated.

Direct cellular verification of EnvZm3t10 phosphatase activity is challenging due to the acid-lability of phosphohistidine and phosphoaspartate bonds[43,44] and lack of commercial antibodies against P-OmpR. To verify that EnvZm3t10 acts as a phosphatase, we thus carried out additional control experiments. Deactivation of OmpR-driven output by EnvZm3t10 is abolished when adding mutations predicted to eliminate its phosphatase activity (m3 – [N343A]), or using constitutively active variants of OmpR-VP64 (Supplementary Figure 8). Thus, the observed putative phosphatase activity is not caused by blocking interactions between the kinase and OmpR-VP64, nor by sequestration of OmpR-VP64. It is thus unlikely that EnvZm3t10 is acting through a mechanism other than direct dephosphorylation of P-OmpR-VP64.

**Tuning kinase-output responses via phosphatase activity**. We next constructed a family of tunable genetic devices in which the tunability arises from a CMC between our preferred kinase (EnvZm2) and phosphatase (EnvZm3t10) acting on OmpR-VP64 (Fig. 4a). The inputs to these devices are the enzymatic activities of the kinase ($u_K$) or phosphatase ($u_P$), or factors that affect such rates. The device outputs are the transcriptional and translational products driven by OmpR-VP64. To evaluate the tunability of our engineered CMC, we compared the level of OmpR-VP64-driven output across combinations of kinase and phosphatase levels, with the phosphatase level modulated at the DNA and protein levels (Fig. 4b–c).

First, we titrated both kinase and phosphatase levels by dosing in different amounts of plasmid DNA per sample using poly-transfection[45] (Fig. 4b). The 2D input–output map shows that output expression increases gradually as the ratio of kinase-to-phosphatase increases (Fig. 4b, left). As the dosage of phosphatase increases, the amount of kinase needed to activate the output increases (Fig. 4b, center), indicating a decreased sensitivity to kinase input levels. Likewise, as the level of kinase increases, the amount of phosphatase needed to deactivate the output also increases (Fig. 4b, right). Both results are in accordance with standard models of CMCs[24] (see Supplementary Note 1 for our derivation).

Following the above results, we predicted that we could tune output expression through the modulation of phosphatase

stability (Fig. 4c). To do so, we fused the phosphatase to either of the small molecule-inducible degradation domains (DDs) DDd[46] and DDe[47], which are stabilized by the addition of trimethoprim (TMP) and 4-hydroxytamoxifen (4-OHT), respectively. N-terminal fusions of both DDd and DDe showed the highest fold-changes in output expression upon addition of the cognate small molecule (Supplementary Figure 9); we chose to move forward with DDd/TMP for further testing due to lower background signal than DDe/4-OHT. Titration of both the kinase dosage and TMP concentration shows that the output is high only when the kinase is high and TMP is low (Fig. 4c, left). The addition of TMP decreases the sensitivity of the output to a kinase (Fig. 4c, center) and the addition of kinase decreases the sensitivity of the output to TMP (Fig. 4c, right).

The response of the TMP-tuned design to kinase and TMP levels depends on the initial level of phosphatase in the cell. If the level of phosphatase is initially too high, the DDs cannot suppress it enough to enable output induction by the kinase; conversely, if the initial level of phosphatase is too low, the kinase dominates the CMC even without any TMP added (Supplementary Figures 10 and 11). Thus, there is an optimal level of phosphatase where TMP-induced deactivation of gene expression is maximized.

**Engineered, cell-type-specific signaling responses**. In addition to ectopically expressed factors, endogenous cellular factors can also be plugged in as inputs to the kinase ($u_K$) and phosphatase ($u_P$) in our engineered CMC, enabling device performance to be tuned based on factors such as the state of the cell. One particularly useful class of intracellular inputs are miRNAs, which are differentially expressed across cell types[48] and can be used to identify specific cell states[49]. Building on our CMC, we expected that endogenous miRNAs can be targeted to the mRNAs of the kinase or phosphatase to decrease or increase output expression, respectively (Fig. 5a). An important and difficult challenge in miRNA sensing is to achieve good on/off responses from the conversion of "high" miRNA inputs into high levels of output expression[45]. We thus investigated our CMC as a scaffold for improving miRNA input processing and generating cell-type-specific signaling responses.

As a proof of concept, we built a sensor for a cancer-associated miRNA, miRNA-21-5p (miR-21), which has previously been used to classify HeLa cells as distinct from HEK cells[45,49]. To do so, we placed four miR-21 target sites (T21) in both the 5′- and 3′-UTRs of the phosphatase transcription unit (Fig. 5b). As a control, we replaced the miR-21 target sites with four target sites for the synthetic miR-FF4 (TFF4)[50]. In cells expressing miR-21, we expected the phosphatase to be knocked down, thereby dramatically shifting the balance of the CMC to favor phosphorylation of OmpR-VP64 and thus activation of the output. Since P-OmpR has only a ~10–30-fold higher affinity for DNA binding compared to OmpR[51] (which we validated in HEK-293FT cells—Supplementary Figure 12), we included an endoribonuclease (endoRNase)-based incoherent feedforward loop (iFFL)[52] to constrain cell-to-cell variance in the expression level of OmpR-VP64 (Supplementary Figure 13). This is helpful due to the high DNA dosage variance of transfections, within which only a small subset of cells typically receive the ideal dosage of OmpR-VP64, and cells that receive high DNA dosages are susceptible to spurious activation of output expression by unphosphorylated OmpR.

To test the circuit, we first considered the effect of miR-21 on the kinase-output dose-response curve. We expected that endogenous expression of miR-21 would selectively sensitize output expression to kinase levels in HeLa cells. Without the phosphatase, the kinase can induce output expression in both

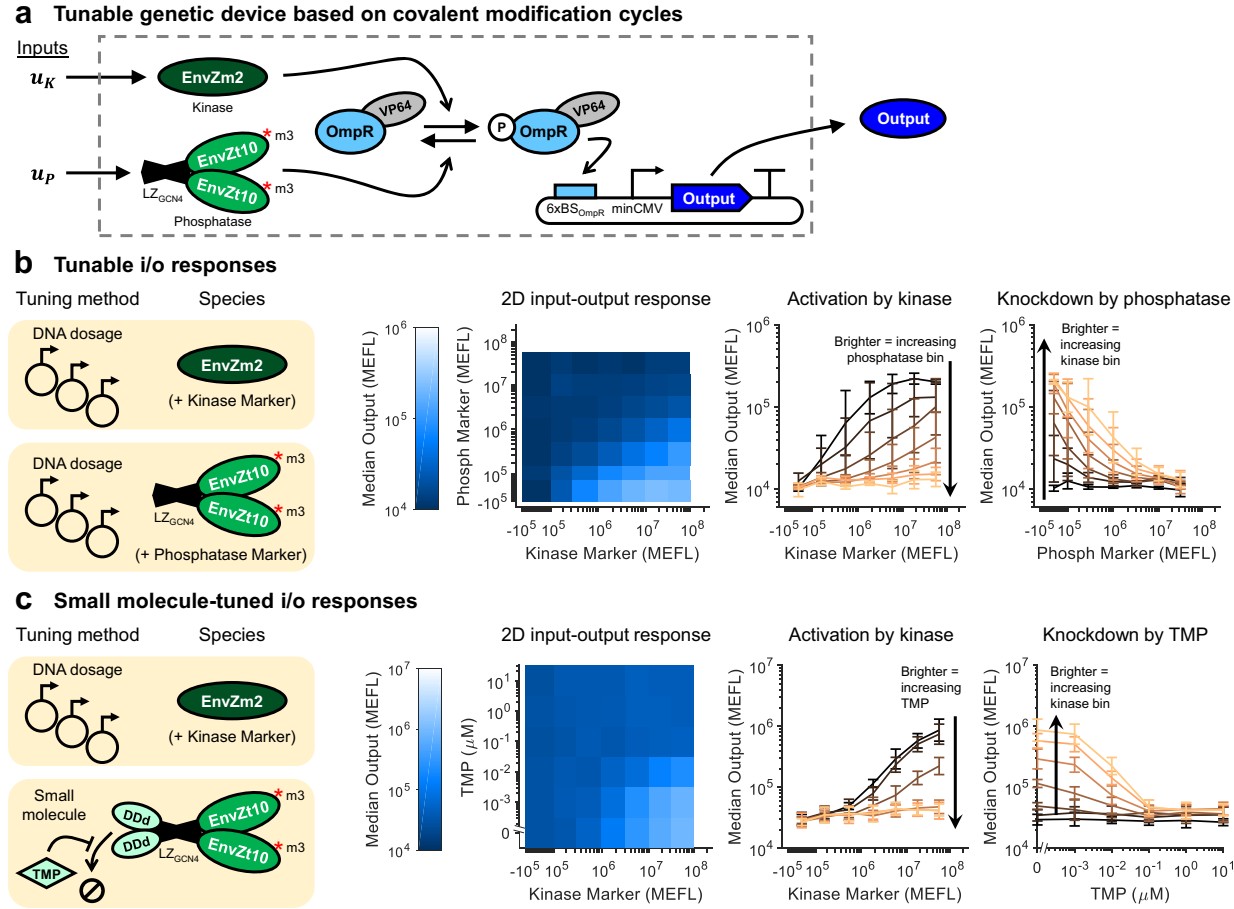

**Fig. 4 Tuning input/output signaling response by modulating kinase and phosphatase levels. a** Implementation of a covalent modification cycle with kinase (EnvZm2) and phosphatase (EnvZm3t10) variants of EnvZ. The expression level of the output can be tuned as a function of both enzymes, and inputs to each that affect their production rate ($u_K$ and $u_P$). **b** Tuning output expression through different dosages of kinase and phosphatase DNA. The heatmap shows the median level of output for each combination of kinase and phosphatase DNA dosages, assayed with poly-transfection[45] (see Supplementary Figure 14 for full data). The line plots show the same data but are broken out by rows or columns. Brighter lines correspond to bins with increasing phosphatase (left) or kinase (right). **c** Tuning output expression through small molecule-induced degradation of the phosphatase. DDd is fused to the N-terminus of the phosphatase (see Supplementary Figure 10 for different arrangements and comparison with DDe/4-OHT). The addition of TMP stabilizes the DDd-phosphatase fusion protein[46]. The data are extracted from the full poly-transfection results shown in Supplementary Figures 10 and 11, selecting the middle phosphatase bin ($P$ Marker $\approx 10^6$). The line plots show the same data but broken out by rows or columns. Brighter lines correspond to samples with increasing TMP concentration (left) or bins with increasing kinase (right). All data were measured by flow cytometry at 48 hours post transfection. HEK-293 cells were used for **b** and HEK-293FT for **c**. All error bars represent the mean ± s.d. of measurements from three experimental repeats. All heatmap values represent the mean of measurements from three experimental repeats. Source data are provided as a Source Data file.

HEK and HeLa cells with either circuit variant (T21 or TFF4), though with stronger output in HeLa cells (Fig. 5c, left). As expected, at high dosages of phosphatase DNA, output induction by the kinase is suppressed by the phosphatase in all cases except in HeLa cells with the T21 circuit variant (Fig. 5c, right). Note that without the iFFL, the output expression has a higher "leaky" background expression at low ratios of kinase-to-phosphatase dosages (Supplementary Figures 14 and 15). Depending on the phosphatase dosage, the T21 variant in HeLa cells has between 10- and 1000-fold higher sensitivity to kinase input than the TFF4 variant (Supplementary Figure 16). Thus, these results illustrate the ability to use miRNA sensors for cell-type-specific tuning of signaling responses.

To optimize our sensor for cell-type classification, we followed the approach of Gam et al.[45] and used poly-transfection to systematically compare the percent of cells positive for output expression at different ratios of each circuit component. In our previous classifier designs, a transcriptional repressor such as LacI[49] or BM3R1[45] is repressed by the miRNA, thereby

de-repressing output transcription. Poly-transfection analysis showed that miRNA sensing in such systems is difficult to optimize, requiring the expression level of the repressor to be not too high to prevent de-repression and not too low to prevent repression in the first place[45]. In our current design, miRNA sensing is instead optimized by the ratio of kinase-to phosphatase activity, which is a more flexible and easily tuned quantity.

We found that a 1:1:0.5 ratio of Kinase:Phosphatase:Output plasmids (the latter of which was co-delivered with the CasE/ OmpR-VP64 iFFL) maximized classification accuracy for the T21 vs TFF4 variants in HeLa cells (Supplementary Figure 17). At this ratio, we obtained a significant ~50 percentage point increase in cells positive for output reporter between circuit variants in HeLa cells and a ~55 percentage point increase between HeLa and HEK-293 for the T21 variant ($p = 0.0017$ and 0.0056, respectively, paired two-tailed Student's $T$ test—Fig. 5d). The area under the curve (AUC) of the ROC curve of the circuit was $0.83 \pm 0.01$ when comparing T21 vs TFF4 variants in HeLa cells and $0.93 \pm 0.01$ when comparing the T21 variant in HEK-293 vs HeLa

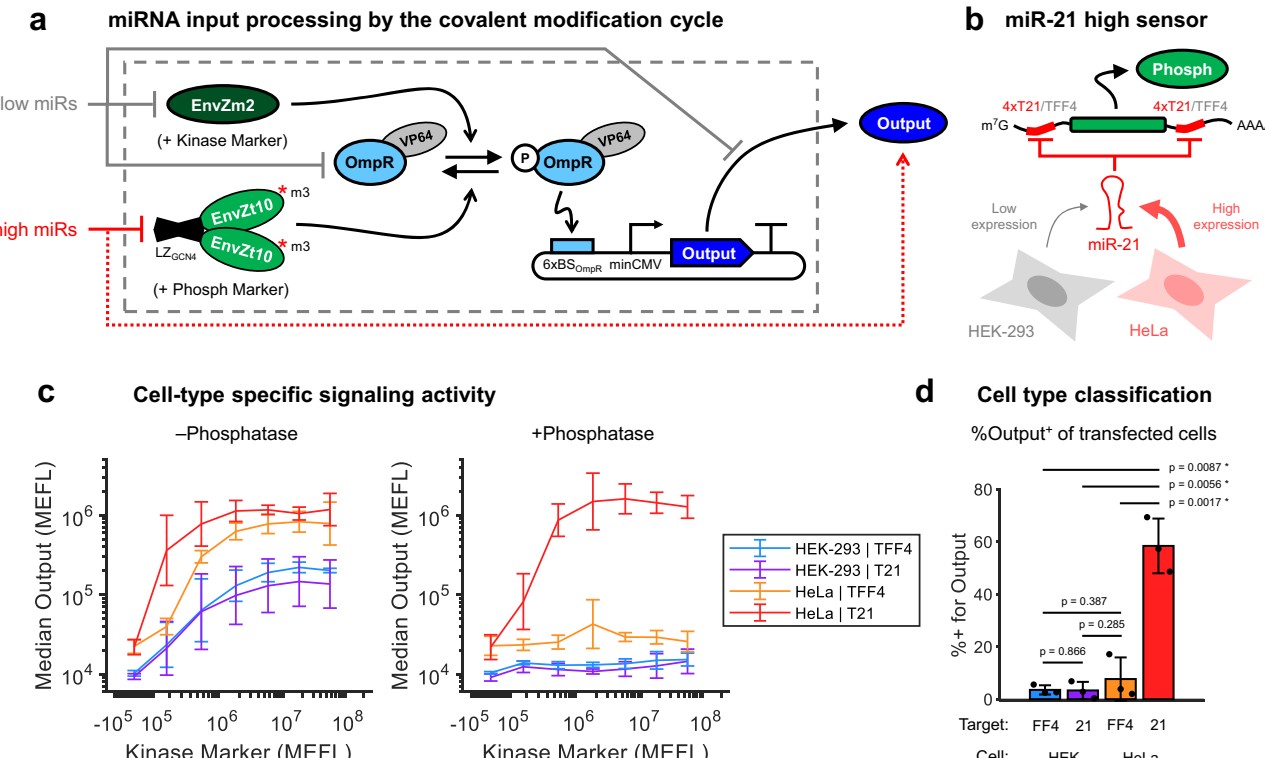

**Fig. 5 Cell type-specific signaling responses using covalent modification cycles. a** miRNA classifier design based on covalent modification cycles. miRNAs expected to be low in the target cell can be used to knock down the kinase, OmpR-VP64, and/or the output. miRNAs expected to be high in the target cell can be used to knock down the phosphatase, effectively increasing the output expression. Not shown for brevity, the level of OmpR-VP64 was optimized using a feedforward controller (Supplementary Figure 13). **b** Design of a miR-21 sensor for classification of HeLa cells. miR-21 knocks down phosphatase levels via 4x target sites in each of its 5' and 3' untranslated regions (UTRs). As a control, a variant was made with miR-FF4 target sites (TFF4) in place of the miR-21 target sites (T21), thus preventing knockdown by miR-21. miR-21 is differentially expressed in HeLa compared with HEK-293 cells[45,49]. **c** Cell type-specific signaling responses are enabled by miRNA regulation of phosphatase expression. The data is extracted from the full poly-transfection data (Supplementary Figure 14), comparing the second-highest phosphatase bin (P Marker ≈ $10^7$) to the lowest (no phosphatase). **d** Comparison of the percent of transfected cells positive for the output for each circuit variant in HEK/HeLa cells at the optimal ratio of kinase to phosphatase markers, extracted from the full poly-transfection data. *P* values are from two-tailed paired T-tests between each group of samples. Receiver-operator characteristic (ROC) curves are provided in Supplementary Figures 17 and 18. All data were measured by flow cytometry at 48 hours post transfection. All error bars represent the mean ± s.d. of measurements from three experimental repeats. Source data are provided as a Source Data file.

cells (Supplementary Figure 17). Examining various combinations of dosages of the kinase, phosphatase, and output reporter, we found that the AUC of the resulting ROC-like curve of our phosphorylation-based classifier (0.93 ± 0.04—Supplementary Figure 18) is higher than that of our recently optimized transcriptional repressor-based classifier (0.84—see SI Fig. 16 in Gam et al.[45]) for discriminating HEK vs HeLa cells, suggesting improved overall performance for cell-type classification. Thus, the CMC can be used for robust miRNA input processing with minimal tuning effort by finding the optimal ratio of kinase-to-phosphatase activities.

**Design of a phosphorylation-based feedback controller.** The responses of expressed genes to their extracellular (or intracellular) inputs are often stochastic and thus imprecise across individual cells[53,54]. In addition, the intracellular context affects the level of gene expression induced by signaling[30,31], owing to factors such as off-target interactions[55] or resource competition[52,56,57] among engineered genes. To remedy these issues and enable the construction of signaling circuits that enforce precise and robust signaling responses across cells, we applied negative feedback control to our CMC (Fig. 6a). In both natural and synthetic systems, feedback control can reduce

cell-to-cell variance of gene expression in response to signal inputs[58–60]. Negative feedback has also been used to make gene expression robust to perturbations that affect processes within the feedback loop[61–63]. An advantage of our controller design is that it can be applied without modifying any promoters or intermediate RNA or protein species in the pathway (e.g., via the generation of fusions), and simply requires a modification of the output mRNA.

In our controller, the phosphatase is co-expressed with the output gene via a 2A linker[64] and suppresses its own production via dephosphorylation of P-OmpR-VP64 (Fig. 6b). Feedback strength can be tuned through TMP regulation of the DDd-fused phosphatase. The level of output ultimately set by the controller thus arises from competitive phosphoregulation of OmpR-VP64 by the kinase and feedback phosphatase. In an ideal system operating with both enzymes saturated, the concentrations of the phosphatase and the output species become insensitive to disturbances affecting their gene expression processes (see Model box). As TMP selectively regulates phosphatase but not output stability, it can be used as an input to the controller to tune the strength of the feedback. Under the ideal conditions presented above and while OmpR-VP64 has not saturated the output promoter, the relationship between the levels of kinase and output is independent of both the exact mechanism by which OmpR-VP64 activates output

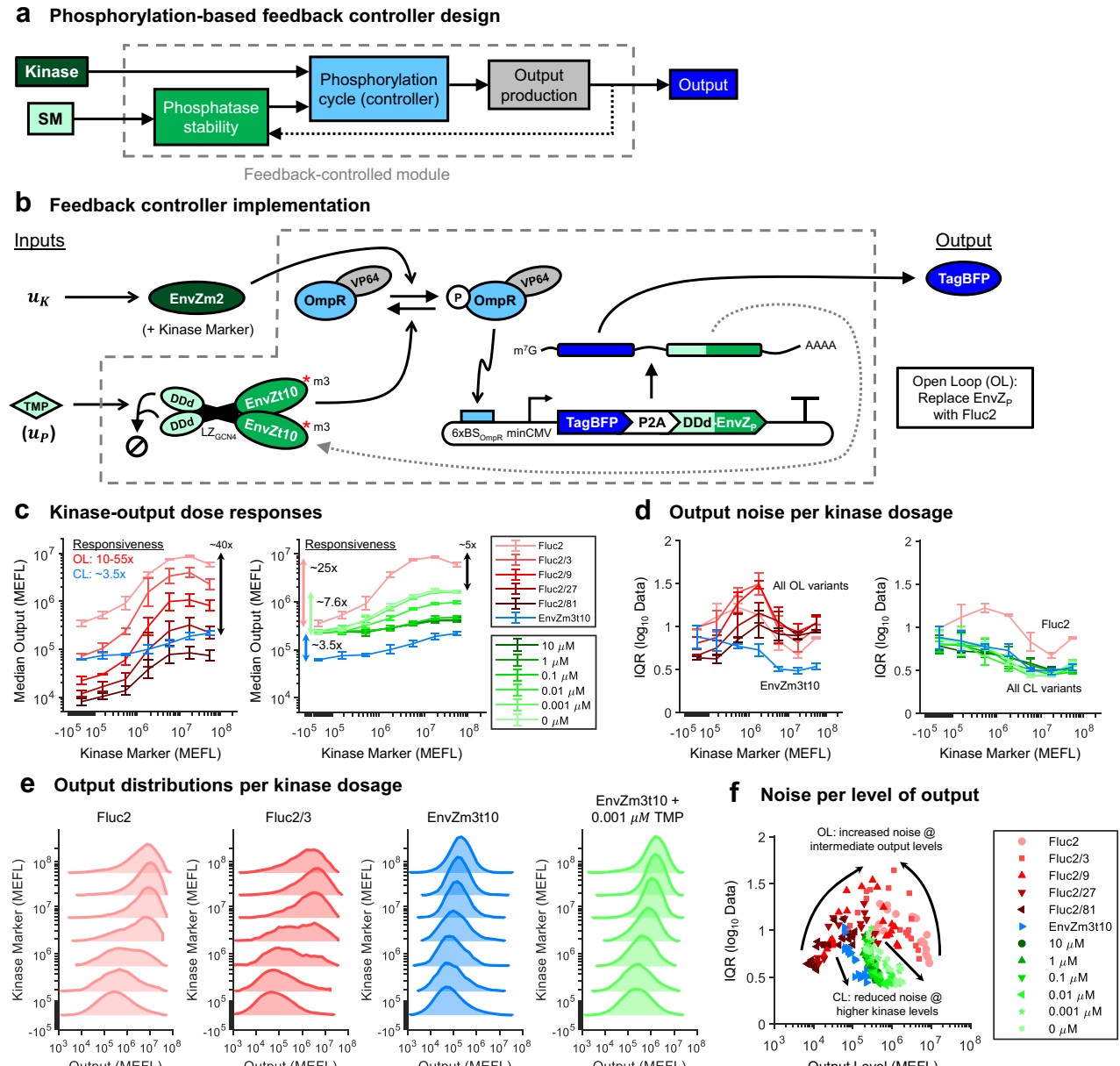

**Fig. 6 Design and implementation of a tunable phosphorylation-based feedback controller. a** Block diagram of the feedback controller design. The phosphatase acts as an output sensor, and is fed back from the output to the phosphorylation cycle of the TF that activates output production. The kinase sets the reference for output expression. The output responds to inputs both to the kinase and a small molecule (SM) regulator of phosphatase stability, the latter effectively serving to tune the feedback strength. **b** Implementation of the feedback controller. The kinase is EnvZm2, the phosphatase is DDd-EnvZm3t10, and the output is the fluorescent reporter TagBFP. The output is 2A-linked[64] to the phosphatase to ensure coupled transcription. The addition of TMP stabilizes DDd-EnvZm3t10 and thereby increases the feedback strength. An open-loop (OL) version of the system was made by replacing the phosphatase with the luminescent protein Fluc2. Since negative feedback reduces output expression, OL variants with reduced output levels were created for comparison at equivalent output levels by reducing the relative copy number of output reporter by fractional amounts (1:3, 1:9, 1:27, and 1:81). The kinase was poly-transfected in a separate complex to the other plasmids to measure the dose–responses of the OL and closed-loop (CL) systems (see Supplementary Figures 19 and 21 for details). **c** Dose–responses of OL and CL system outputs to kinase input levels. The range of responsiveness to a kinase (max fold change ± kinase) is given for the CL and OL variants are indicated to the left of the lines. The fold-difference between max output levels for select OL and CL variants are indicated to the right of the lines. Dose–responses of the DDd CL system to TMP input are given in Supplementary Figure 24. **d** Quantification of output noise as a function of kinase input dosages. As the output variance is log-distributed, the interquartile range (IQR) is computed on the log-transformed data. **e** Comparison of output distributions for select OL and CL variants across kinase levels. The data are representative of the first experimental repeat. All OL and CL variants are compared in Supplementary Figure 23. **f** Noise as a function of median output levels for all CL and OL variants at all kinase inputs. The individual points are drawn from all experimental repeats. All data were measured by flow cytometry at 48 hours post transfection in HEK-293FT cells. All error bars represent the mean ± standard deviation of measurements from three experimental repeats. Source data are provided as a Source Data file.

expression as well as any perturbations in the transcription and translation processes of the output/phosphatase (see Model box).

To evaluate the performance of the feedback controller, we first measured the kinase-output responses for open loop (OL) and closed loop (CL) variants. The OL system was made by replacing the phosphatase with Fluc2, which has no effect on OmpR phosphorylation (Fig. 2b, c). Since the presence of negative feedback reduces the level of output expression for a given input level of kinase, we tested several OL variants in which the amount of output reporter plasmid in transfections was reduced by 3×, 9×, 27×, or 81× (respectively, referred to as Fluc2/3, Fluc2/9, Fluc2/27, and Fluc2/81). We define kinase responsiveness as the maximal output fold-change in the presence versus absence of kinase. The kinase responsiveness of the OL systems varies from ~10-to-55-fold, whereas for the CL system variant without DDd fused to the phosphatase, it is ~3.5-fold (Fig. 6c, left – see Supplementary Figures 19-22 for full poly-transfection scheme and data). Adding DDd to the phosphatase increases the CL kinase responsiveness to ~7.6-fold without TMP, and ~6.4-fold for the lowest non-zero amount of TMP that we tested: 0.001 μM (Fig. 6c, right). The kinase responsiveness of the CL system decreases as more TMP is added (and thus the phosphatase is stabilized) to the point of approximately matching that of the non-DDd CL system (Fig. 6c, right). The maximum output level of the DDd CL system is up to 10-fold higher than that of the non-DDd CL system and within ~5-fold that of the OL system. Thus, tuning the feedback strength via TMP allows the CL system to recover approximately one-third of the dynamic range of the OL system.

We next compared variance in circuit output levels across cells. In the absence of kinase input, we see similar levels of noise in output expression for all OL and CL variants; however, as the dosage of the kinase is increased, we observe a decrease in noise for CL variants and an increase in noise for OL variants (Fig. 6d). At high dosages of kinase, the output noise for OL devices decreases again but does not reach the low noise achieved in CL devices. The higher noise in OL systems can be attributed to a more digital-like transition in output expression per cell as the kinase dosage increases, whereas in CL systems we observe a smooth, unimodal shift in output expression per cell (Fig. 6e, see Supplementary Figure 23 for all variants). The decrease in noise in CL output expression as a function of increasing kinase can likely be attributed to the increasing concentration of P-OmpR-VP64, on which the phosphatase can actuate negative feedback. Interestingly, tuning feedback strength via TMP appears to have little effect on the magnitude of output noise (Supplementary Figure 24), suggesting that the faster degradation of the phosphatase did not push our system into a regime where the negative feedback is substantially attenuated.

Comparing the noise as a function of output level for all CL and OL variants tested, we can see that the noise in the OL system peaks at intermediate absolute levels of output (regardless of the kinase dosage needed to achieve such an output level for a given OL variant), whereas the noise in the CL systems decreases as the output increases due to the factors described above (Fig. 6f). The pattern of noise in the OL variants can potentially be explained by stochastic transcriptional variation among cells when the output promoter is not saturated. Through negative feedback, the CL system is likely able to suppress this source of noise.

**Robustness to perturbations via feedback control**. According to our mathematical modeling comparing the OL and CL circuits, the presence of negative feedback is expected to impart robustness to perturbations that affect the expression of the output protein (see Model box). We analyzed robustness in terms of both

---

### Model box

Here, we develop a mathematical model to show that CMC-mediated feedback enables the expression level of a regulated gene to be robust to disturbances. In particular, for a fixed kinase level ($K_t$), we treat the genetic circuit shown in Fig. 7 as feedback interconnection of two dynamical processes with input/output (i/o): an engineered CMC that takes phosphatase concentration ($P_t$) as input and outputs P-OmpR-VP64 concentration ($X^*$), and a gene expression process that takes $X^*$ as input to produce the phosphatase $P_t$ as output. We use a standard Goldbeter-Koshland model[24] for the dynamics of the CMC:

$$\frac{d}{dt}X^* = \theta_k \frac{(X_t - X^*)K_t}{(X_t - X^*) + K_{M,k}} - \theta_p \frac{X^*P_t}{X^* + K_{M,p}}, \quad (1)$$

where $\theta_k$ and $\theta_p$ are catalytic rate constants of the kinase and the phosphatase, respectively, $K_{M,k}$ and $K_{M,p}$ are their respective Michaelis–Menten constants, and $X_t$ is the total amount of OmpR-VP64 (i.e., OmpR-VP and P-OmpR-VP). The expression of $P_t$ is regulated by an OmpR-activated promoter, which gives rise to the following dynamics:

$$\frac{d}{dt}P_t = \alpha(1 - w)\phi(X^*) - \gamma P_t, \quad (2)$$

where $\alpha$ is the production rate of $P_t$ that lumps the rate constants for transcription, translation, and mRNA decay, $\phi(\cdot)$ is a Hill function satisfying $\phi' > 0$ for all $X^*$, $\gamma$ is the protein decay rate constant, and $0 \leq w < 1$ is a disturbance that models the fold change in the production rate of $P_t$, which could either arise from indirect transcriptional repression via resource loading or from direct post-transcriptional repression via miRNA (see Fig. 7). The output from this feedback-regulated gene is $Y = \rho P_t$, since the output protein and phosphatase are co-transcribed but produced as separate proteins using a 2A-linker. We find that the relative sensitivity of output to disturbance $w$ for this closed-loop system (1–2) at a given output level $Y$ is

$$\mathcal{S}_{CL}(Y) = \frac{1}{Y} \cdot \left|\frac{dY}{dw}\right| = \frac{1}{1-w}\left[1 + \frac{\alpha}{\gamma}(1-w)\left|\frac{d}{dY}(\phi \circ h)\right|\right]^{-1}, \quad (3)$$

where $h$ is the transfer curve of the CMC. In comparison, when the CMC in (1) is not connected with (2), the relative sensitivity of $y$ to disturbance $w$ for the open-loop system (2) is $\mathcal{S}_{OL} = \frac{1}{1-w}$. Hence, we have $\mathcal{S}_{CL} < \mathcal{S}_{OL}$ for all $y$ regardless of where the sensitivity is evaluated. This implies that the closed-loop system is always more robust than the open-loop system to disturbance $w$. To enable near-perfect adaptation to $w$, it is sufficient to increase $T := \left|\frac{d}{dy}(\phi \circ h)\right| = |h' \cdot \phi'|$. In particular, if $T \to \infty$, then $\mathcal{S}_{CL} \to 0$, implying that the closed-loop system can perfectly adapt to $w$. Specifically, for any fixed $X^*$ and $y$, there exists sufficiently small $K_{M,p}$ and sufficiently large $X_t$ to make $|h'|$ arbitrarily large. On the other hand, to ensure $T$ is large, $|\phi'|$ must not be too small. This requires us to design the system to prevent saturation of the OmpR-activated promoter. Promoter saturation limits the ability of the output to respond to changes in OmpR phosphorylation, and thus can limit the benefit of the negative feedback to achieve robustness to perturbations. Hence, the KD of binding between phosphorylated OmpR and its target promoter must not be too small[28]. Under the ideal operating conditions described above, both enzymes are saturated by their substrates, which is possible for a small $K_{M,p}$ and large $X_t$. Specifically, if $K_{M,p} \ll X^*$ and $X_t \gg K_{M,k}$, equation (1) can be approximated by $dX^*/dt = \theta_k K_t - \theta_p Y/\rho$, leading to quasi-integral feedback control[28].

---

fold-changes in gene expression resulting from the perturbations and a robustness score (100% minus the percent deviation from the unperturbed level); a high degree of robustness is indicated by a small absolute fold-change and a high robustness score. We tested the capability of the CL system to impart robustness of output expression levels to perturbations that model off-target gene regulation and resource loading (Fig. 7a). To model off-target regulation by an endogenously- or ectopically-expressed

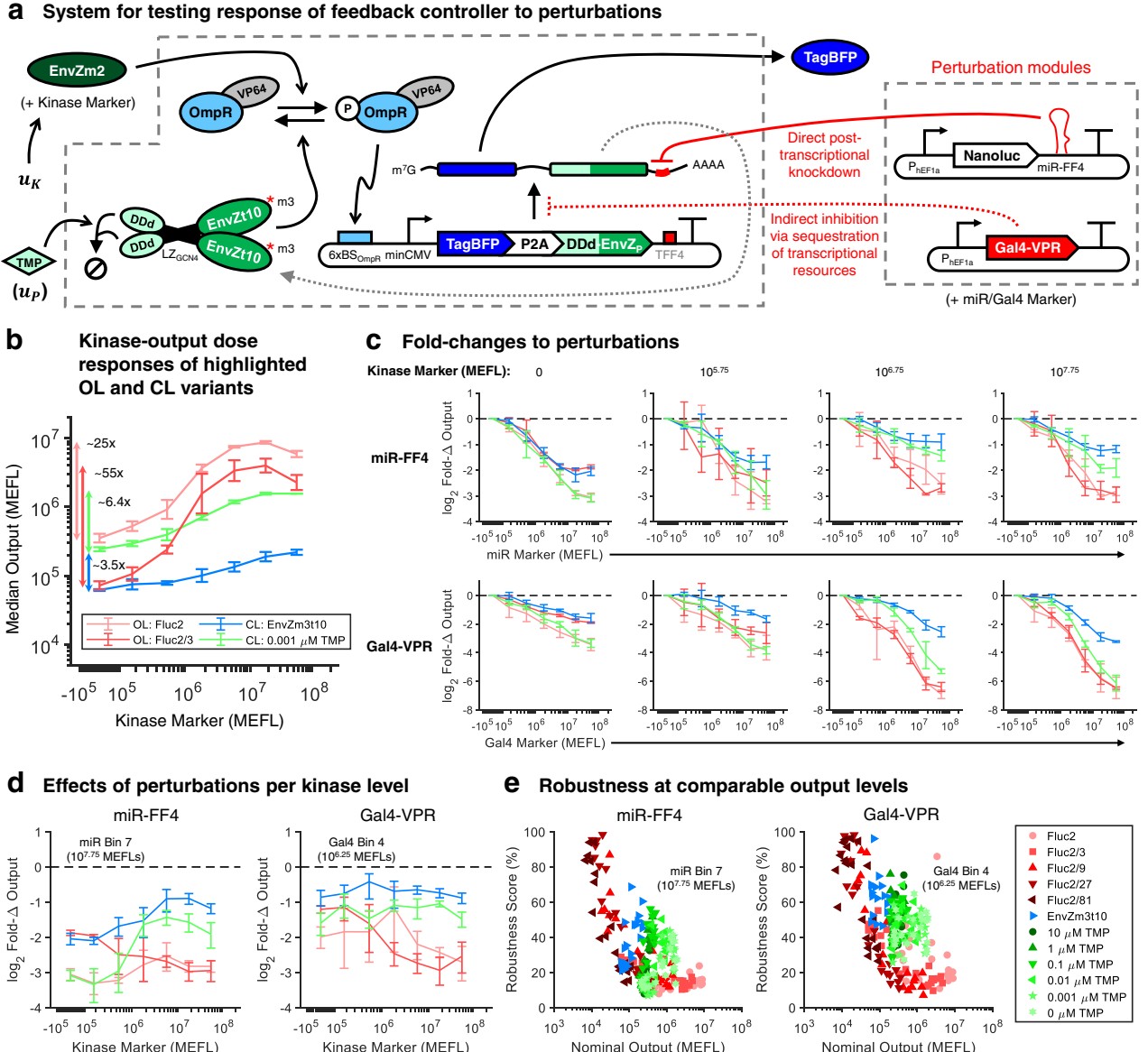

**Fig. 7 Mitigation of perturbations via feedback control. a** The CL and OL systems introduced in Fig. 6 were tested against two perturbations: (i) indirect transcriptional inhibition via loading of transcriptional resources by Gal4-VPR and (ii) direct post-transcriptional knockdown by miR-FF4. The kinase, perturbations, and controllers were each poly-transfected in separate DNA-lipid complexes in order to measure 2D dose-response of the OL and CL systems to the kinase and perturbations (see Supplementary Figures 19–22 for details). **b** Dose–responses of OL and CL systems highlighted in the following panels. The Fluc2 and Fluc2/3 OL variants were chosen since they have nearly identical output levels compared to the CL with and without DDd, respectively, in the absence of kinase. Dose–responses and detailed comparisons among all OL and CL variants are provided in Supplementary Figures 25–28. **c** Fold-changes (Fold-Δs) in output expression in response to miR-FF4 (top row) and Gal4-VPR (bottom row) perturbations. Each column represents an increasing amount of kinase input from left to right. The dashed lines indicate no fold-change (ideal). **d** Direct comparison of fold-changes to perturbations between OL and CL variants across kinase dosages. The data represents the maximum dosage of miR-FF4 (miR Marker $\approx 10^{7.75}$ MEFLs) and dosage of Gal4-VPR with a comparable level of knockdown to the OL (Gal4 Marker $\approx 10^{6.25}$ MEFLs). **e** Robustness scores (100% − % deviation due to perturbations) for all OL and CL variants across each kinase input level at the same dosages of miR-FF4 and Gal4-VPR as highlighted in **d**. Nominal outputs indicate the level of output in the absence of any perturbations. The individual points are drawn from all experimental repeats. All data were measured by flow cytometry at 48 hours post transfection in HEK-293FT cells. All error bars represent the mean ± standard deviation of measurements from three experimental repeats. Source data are provided as a Source Data file.

gene regulator such as a miRNA, we expressed miR-FF4, which binds and cleaves a target site (TFF4) placed in the 3′-UTR of the output/phosphatase mRNA, causing mRNA degradation. To model resource loading, we expressed Gal4-VPR, which strongly sequesters transcriptional resources, such as those recruited by the VP64 activation domain fused to OmpR, reducing transcription of other genes[52]. In addition to the modeled effects, these perturbations are useful because they affect output

production both before (Gal4-VPR) and after (miR-FF4) transcription, enabling comparison of the CL system's ability to respond to perturbations at different stages of gene expression.

As expected, we found that the CL system is indeed more robust than comparable OL systems to miR-FF4 and Gal4-VPR perturbations (Fig. 7b–e). Detailed comparisons of the response of all OL and CL variants to both perturbations are provided in Supplementary Figures 25 and 26. For illustration, we highlight

and compare two OL and two CL variants with similar basal output levels in the absence of kinase: Fluc2, Fluc2/3, EnvZm3t10, and DDd-EnvZm3t10 + 0.001 μM TMP (Fig. 7b). Without kinase, there is little difference between the effects of miR-FF4 and Gal4-VPR on the OL and CL systems (Fig. 7c, left panels), consistent with the expected lack of feedback actuation in the absence of P-OmpR-VP64 and our earlier findings of similar levels of noise in the same regime (Fig. 6d). At higher kinase input levels, the fold-changes in output expression in response to both perturbations are substantially smaller for the CL variants than the OL variants (Fig. 7c, right panels).

The relative decrease in fold-changes as a function of kinase input dosage is plotted in Fig. 7d for two levels of miR-FF4 and Gal4-VPR perturbations that knock down the OL systems to similar degrees. At medium-to-high kinase input levels, the feedback controller can respond to the perturbations by sustaining the output level to within 2–4-fold of the nominal (unperturbed) levels, improving significantly over the 6–10-fold changes observed in the OL systems. The relatively weaker output suppression by Gal4-VPR for both the OL and CL variants at low-kinase dosages may result from generally weaker effects of transcriptional resource sequestration on basal transcription vs activated transcription[65]. This may offset the generally increased susceptibility of the CL system to perturbations in the low-kinase regime, causing the CL systems to be more evenly perturbed by Gal4-VPR across kinase dosages.

Because negative feedback reduces output expression, and since both miR-FF4 and Gal4-VPR knock down gene expression, a full comparison of the effects of these perturbations on the OL and CL systems must account for differences in the nominal output expression level. This is because lower nominal output levels have a reduced measurable dynamic range of knockdown due to detection limits imposed by the autofluorescence background. To account for varying nominal output levels for OL and CL systems at different kinase input levels, we compared the nominal output level versus robustness score for each device. Collating all CL and OL variants at the same miR-FF4 and Gal4-VPR dosages as in Fig. 7d, we can see that the CL systems are nearly always more robust than the OL systems for a given nominal output level (Fig. 7e). The only substantial overlap in the plots between the OL and CL systems occurs at low-kinase inputs to the CL system. Quantitatively, for a given nominal output level, we see a 20–30 percentage point increase in robustness score for the CL systems compared with the OL variants. Comparisons across additional dosages of each perturbation show similar results (Supplementary Figures 27 and 28). Thus, our phosphorylation-based feedback controller is capable of reducing the impact of perturbations on the expression of the output gene at both the transcriptional and post-transcriptional levels. Coupled with the reduction in noise (Fig. 6), these data indicate that the feedback controller can successfully impart precise, tunable, and robust control over gene expression in mammalian cells.

## Discussion
Here, we developed tunable and precise signaling circuits in mammalian cells that are robust to perturbations using engineered CMCs derived from bacterial TCS proteins (Fig. 1). We first screened engineered variants of the *E. coli HK* EnvZ to isolate kinase and phosphatase activity from this bifunctional protein (Figs. 2 and 3). We demonstrated tunability in kinase-induced gene expression responses conferred by small molecule-inducible expression of a phosphatase (Fig. 4). Building upon this tunability, we showed that incorporating target sites for endogenous miRNAs can be used to create cell type-specific signaling responses through knockdown of phosphatase expression.

Co-expressing the phosphatase with the output, we created a tunable negative feedback loop that reduces both cell-to-cell variation and sensitivity to perturbations of kinase-induced gene expression (Figs. 6 and 7).

Combined with recent advances in utilizing TCS proteins to engineer synthetic receptors in mammalian cells[13,14] and to rewire the specificity of RRs in bacteria[66], our platform will enable the construction of sophisticated synthetic signaling systems that can connect intracellular and extracellular inputs to diverse target output in mammalian cells. Although much work has so far focused on synthetic receptor engineering[10,11], incorporation of downstream signal processing moieties to improve signaling pathway function has only recently begun to be explored[67]. In particular, the ability to easily tune signaling pathway activity through phosphatase expression and the ability to robustly control downstream gene expression processes will facilitate the creation of synthetic signaling systems that can operate across diverse cellular contexts. In the future, our circuits can form the basis for advanced cellular computing[68] and feedback control[69] architectures. In addition, the phosphorylation cycles in our CMC-based systems may serve to effectively buffer the effects of retroactivity[70] from downstream target sites loading the phospho-TF, provided that the phosphorylation reactions are sufficiently fast and the enzyme concentrations are sufficiently high[71]. Utilizing TCS components comprising multiple His-Asp phosphorelays[19] may further buffer the effects of retroactivity[71,72]. Finally, connecting signaling pathway activity to endogenous gene regulation, such as through miRNA regulation of pathway components, will bolster applications in guiding differentiation or programming custom signaling for different cellular states.

The high degree of orthogonality among existing TCS pathways[73–75] and the relative ease of finding new orthogonal HK/RR pairs[76] indicates that TCS pathways will be a bountiful source of orthogonal signaling pathways for use in mammalian cells. To support this effort, we identified several HK-RR pairs that show good orthogonality in mammalian cells (Supplementary Figures 29–31). Though TCS pathways are absent in animals[15], histidine and aspartate phosphorylation are more prevalent than previously thought[44]. The lack of observed histidine to aspartate phosphotransfer in animals indicates a strong likelihood of orthogonality between TCS pathways and existing signaling networks in animal cells, though future work will be needed to examine possible cross-talk.

Through the implementation of feedback control via CMCs, we have opened the door to creating increasingly precise and robust responses in engineered signaling pathways. Reducing cell-to-cell variation in signaling output is critical for ensuring that cells in a population make uniform, rather than multimodal or stochastic, decisions. Reducing the sensitivity of output expression to perturbations will help further control individual cellular decision-making and ensure that engineered signaling systems can operate across diverse cell types and states[52]. In the future, it may be possible to improve the robustness to perturbations conferred by our feedback controller. To achieve near-perfect adaptation to perturbations, the system parameters need to be tuned such that it can operate as a quasi-integral feedback controller[28,77]. We suspect that the $K_M$ of the phosphatase is likely similar to or higher than the $K_D$ of P-OmpR, reducing the efficacy of the feedback (see Model box). Increasing $\frac{K_D}{K_M}$ helps ensure that the phosphatase is saturated with P-OmpR while the output promoter is not, both of which are critical conditions for the feedback to work in a quasi-integral manner[28] (see Model box). Further discussion of possible future approaches to achieve quasi-integral feedback control with our system are discussed in Supplementary Note 2.

In addition, to set point regulation, negative feedback may also speed up the dynamics of a regulated system, as has been

observed previously using transcriptional negative feedback[78]. However, the faster dynamics of the regulated system depend on controller architecture and parameters. In particular, whereas controllers incorporating integral action are known to improve steady-state regulation performance, they may not speed up dynamics[79], which is why integral control is usually combined with proportional and derivative control in engineering[80]. In terms of analyzing the dynamics of our system, it is important to consider that it was tested with transient transfections, which places limits on measurement fidelity as well as operational responses. DNA dilution during transient transfection prevents the system from achieving a non-zero steady-state, and also causes the setpoint to change over time as the kinase level increases then decreases. Further, DNA dilution causes controller performance to degrade as the expression of OmpR decays and the enzymes begin to operate outside the saturation regime needed for integral control (see Model box). Thus, to precisely evaluate the dynamic response of our controllers and any modified variants thereof, future studies will benefit from genomic integration of the circuits.

In natural systems, feedback control plays a critical role in regulating signaling pathway activities. Both negative and positive feedback is common in TCS pathways[81]. As with the robustness to perturbations conferred by our feedback controller, negative feedback in natural and engineered TCS pathways in bacteria also allows for adaptation to signal inputs[26,73,81,82]. A conceptually similar controller to our design is found in bacterial chemotaxis, in which feedback control via reversible methylation of the receptor protein Tar enables near-perfect adaptation of flagellar motion to chemoattractants[77,83]. Another close analog can be found in the human ERK1/2 MAPK (mitogen-activated protein kinase) pathway[84]. In this pathway, Mek is analogous to our HK kinase, Erk is analogous to OmpR-VP64 (though Erk itself only indirectly activates transcription through its targets[85,86]), and the Erk-induced phosphatases DUSP5/6 are analogous to our HK phosphatase[84,87]. It has been observed that the expression levels of DUSP5/6 are unaffected by ERK1/2 knockdown[88], which we propose may result from adaptation of DUSP5/6 levels to ERK1/2 levels via the negative feedback loop. Negative feedback in both natural and engineered systems, including the ERK1/2 MAPK pathway, has been shown to convert digital, multimodal input-output responses to more graded, linear, and uniform responses[58,59,89]. Likewise, our feedback controller is capable of imparting graded, uniform activation of gene expression in the cell population. Overall, these examples highlight how feedback control plays an important role in the functions of natural systems and will thus serve as a key building block for future synthetic signaling pathways.

In addition to feedback control, natural signaling pathways also incorporate constitutive phosphatases (and regulators thereof) to tune signaling functions across diverse cell types. For example, signaling through the T-cell receptor (TCR) is regulated by several inhibitory receptors such as CD45 and phosphatases such as PTPN22, which suppress TCR pathway activation unless sufficiently high stimulus is encountered[90]. In developing thymocytes, miR-181a-5p suppresses expression of PTPN22, allowing TCR pathway stimulation at lower antigen affinities and providing critical signals for survival and development towards mature T cells[91,92]. In mature T cells, a variety of miRNAs regulate TCR signaling, other signaling pathways, the cell cycle, and secretion, thereby tuning the immunological responses of T cells in response to their environments[93]. Thus, tunable phosphatases and miRNA-regulated signaling responses similar to the ones we developed can be powerful tools for achieving stage-specific control of differentiation and tuning cell behavior in different contexts. Future designs may also incorporate miRNAs that

regulate kinase expression to provide an additional layer of tunability, for example by miRNAs that are lower in cell types or states where higher signaling strengths are desired.

As synthetic biology progresses, the development of artificial signaling pathways that reflect natural pathways through the incorporation of multiple layers of negative feedback and tuning will facilitate increasingly sophisticated and robust control of cellular behavior. The customizable signaling responses enabled through platforms such as ours may be combined with engineered receptors[13,14] and modular effectors[66] to engineer signaling pathways that transmute extracellular inputs to various intracellular functions in mammalian cells. Such engineered signaling pathways will enable precise cell-cell communication and environmental sensing, with applications in engineering cell therapies, scaling up bioproduction, and programming development of stem cells into specific cells, tissues, and organoids.

## Methods

**Modular plasmid cloning scheme.** Plasmids were constructed using a modular Golden Gate strategy similar to previous work in our lab[45,94]. Briefly, basic parts ("Level 0s" [pL0s]—insulators, promoters, 5′-UTRs, coding sequences, 3′-UTRs, and terminators) were created via standard cloning techniques. Typically, pL0s were generated via PCR (Q5 and OneTaq hot-start polymerases, New England BioLabs (NEB)) followed by In-Fusion (Takara Bio) or direct synthesis of shorter inserts followed by ligation into pL0 backbones. Oligonucleotides were synthesized by Integrated DNA Technologies (IDT) or SGI-DNA. pL0s were assembled into transcription units (TUs—"Level 1s" [pL1s]) using BsaI Golden Gate reactions (10–50 cycles between 16°C and 37 °C, T4 DNA ligase). TUs were assembled into multi-TU plasmids ("Level 2s" [pL2s]) using SapI Golden Gate reactions. All restriction enzymes and T4 ligase were obtained from NEB. Plasmids were transformed into Stellar E. coli competent cells (Takara Bio). Transformed Stellar cells were plated on LB agar (VWR) and propagated in TB media (Sigma-Aldrich). Carbenicillin (100l μg/mL), kanamycin (50 μg/mL), and/or spectinomycin (100 μg/mL) were added to the plates or media in accordance with the resistance gene(s) on each plasmid. All plasmids were extracted from cells with QIAprep Spin Miniprep and QIAGEN Plasmid Plus Midiprep Kits. Plasmid sequences were verified by Sanger sequencing at Quintara Biosciences. Genbank files for each plasmid and vector backbone used in this study are provided in Supplementary Data. Plasmid sequences were created and annotated using Geneious (Biomatters).

In addition to the above, we devised a new scheme for engineering synthetic promoters using what we call "Level Sub-0" (pSub0) plasmids. The approach for creating promoters from pSub0 vectors is illustrated in Figure 32. In this system, promoters are divided into up to 10 pSub0 fragments. Because the core elements of a promoter are typically at the 3′-end, we made the pSub0 position vectors start with the 3'-most element and move towards the 5′ of the promoter. Promoter position 1 (pP1) contains the transcription start site, the +1 position for transcription initiation, and surrounding sequences. pP1 can also optionally contain transcriptional repressor binding sites (not done in this study). pP2 contains the TATA box and other upstream core promoter elements[95–97] as desired. Many of the pP1 and pP2 sequences were derived from the minimal promoters studied by Ede et al.[98]. Because the spacing between the TATA box and +1 site is critical[99], we broke apart each minimal promoter at equivalent positions such that they can be interchanged. pP1 and pP2 parts were generally created via PCR reactions using the base pSub0 backbone as a template and adding the inserts via primer overhangs and In-Fusion cloning. Positions 3–10 (pP3–10) are "enhancer" positions, wherein we generally encode binding sites (i.e., response elements) for transcriptional activators (such as the RRs in this study), or enhancers from constitutive promoters (not done in this study). pSub0 plasmids were made by directly ligating annealed primers into pSub0 pP3–10 backbones or through PCR followed by In-Fusion. The annealed primers were synthesized with 4 bp offsets at each end to naturally create overhangs when annealed. All pSub0 plasmids include BsaI binding sites in an analogous position to pL0s, such that pSub0 can be used directly in place of pL0s when generating pL1s (the overhangs are compatible for up to four pSub0 inserts, see Supplementary Table 1). Because pSub0s and pL0s use BsaI for cloning, in the same way, insertion into pL0 backbones using BsaI Golden Gate is inefficient. To more efficiently clone pSub0s into pL0 P.2 (level 0 promoter) plasmids, we thus generally first performed a Golden Gate reaction with the pSub0s separately from the pL0 backbone, then ligated the Golden Gate product with a pre-fragmented and gel-extracted pL0 backbone.

**Cell culture.** HEK-293 cells (ATCC, CRL-1573), HEK-293FT cells (Thermo Fisher, R70007), and HeLa cells (ATCC, CCL-2) were maintained in DMEM containing 4.5 g/L glucose, L-glutamine, and sodium pyruvate (Corning) supplemented with 10% fetal bovine serum (FBS, from VWR). All cell lines used in the study were

grown in a humidified incubator at 37 °C and 5% $CO_2$. All cell lines tested negative for mycoplasma.

**Transfections**. Cells were cultured to 90% confluency on the day of transfection, trypsinized, and added to new plates simultaneously with the addition of plasmid-transfection reagent mixtures (reverse transfection). Transfections were performed in 384-, 96-, 24-, or 6-well pre-treated tissue culture plates (Costar). Following are the volumes, number of cells, and concentrations of reagents used for 96-well transfections; for 384-, 24- and 6-well transfections, all values were scaled by a factor of 0.2, 5, or 30, respectively. In all, 120 ng total DNA was diluted into 10 µL Opti-MEM (Gibco) and lightly vortexed. For poly-transfection experiments, the DNA dosage was subdivided equally among each complex (e.g., for two complexes, we delivered 60 ng DNA in each, 40 ng for three complexes, etc.) The transfection reagent was then added and samples were lightly vortexed again. The DNA-reagent mixtures were incubated for 10–30 mins whereas cells were trypsinized and counted. After depositing the transfection mixtures into appropriate wells, 40,000 HEK-293, 40,000 HEK-293FT, or 10,000 HeLa cells suspended in 100 µL media were added. The reagent used in each experiment along with plasmid quantities per sample and other experimental details is provided in Supplementary Data. Lipofectamine 3000 was used at a ratio of 2 µL P3000 and 2 µL Lipo 300 per 1 µg DNA. PEI MAX (Polysciences VWR) was used at a ratio of 3 µL PEI per 1 µg DNA. FuGENE6 (Promega) was used at a ratio of 3 µL FuGENE6 per 1 µg DNA. Viafect (Promega) was used at a ratio of 3 µL Viafect per 1 µg DNA. The media of the transfected cells was not replaced between transfection and data collection. For all transfections with TMP (Sigma-Aldrich) or 4-OHT (Sigma-Aldrich), the small molecules were added concurrently with transfection complexes. In each transfection reagent-DNA complex, we included a hEF1a-driven transfection marker to indicate the dosage of DNA delivered to each cell.

**Luciferase assays and analysis**. To measure RR-driven luminescence output in Supplementary Figure 29, we used the Promega Nano-Glo Dual-Luciferase Reporter Assay System, following the manufacturer's instructions. In brief, 6000 HEK-293FT cells were transfected using the FuGENE6 reagent with 25 ng total DNA comprising the plasmids hPGK:Fluc2 (pGL4.53), an hEF1a-driven HK, an hEF1a-driven RR, an RR-driven promoter expressing NanoLuc, and filler DNA at 5 ng each. The cells were cultured in 20 µL DMEM supplemented with 10% FBS in 384-well plates with solid white walls and bottoms (Thermo Fisher) to facilitate luminescence measurements. 48 hours post-transfection, cells were removed from the incubator and allowed to cool to room temperature. In all, 20 µL of ONE-Glo EX Reagent was added directly to the cultures, and cells were incubated for 3 mins on an orbital shaker at 900 revolutions per minute (RPM). Fluc2 signal was measured on a BioTek Synergy H1 hybrid reader, with an integration time of 1 s. 20 µL of NanoDLR Stop & Glo Reagent was then added, and cells were again incubated for 3 mins on an orbital shaker at 900 RPM. After waiting an additional 10 mins following shaking, NanoLuc signal was measured on the same BioTek plate reader, with an integration time of 1 s. NanoLuc signals were normalized by dividing by the Fluc2 signals, thereby accounting for differences in transfection efficiency among wells.

**Identification of optimal orthogonal TCS pairs**. To identify the optimal set of orthogonal TCS interactions, we ran a MATLAB script to score all possible combinations of 4–7 HK-RR protein pairs. The script uses a scoring function to evaluate each particular subset of HKs and RRs. The data input into the scoring function is a matrix of output expression levels driven by the RRs in the presence of the selected HKs. The scoring function first identifies a reference value for each row and column by iteratively finding the maximum value in the matrix, blocking off the rest of the values in its row and column, then repeating until each row and column has one reference value. The reference value is then divided by the rest of the values in its row and column, and the quotients are multiplied together to give a score. The scores for each reference value are then again multiplied together to get a final score for a particular combination of HKs and RRs. After iterating through all possible such combinations, the highest final score for a given submatrix size is selected. The method gave qualitatively orthogonal combinations for up to seven TCS pairs; we thus present the optimized seven-matrix in Supplementary Figure 29.

**Flow cytometry**. To prepare samples in 96-well plates for flow cytometry, the following procedure was followed: media was aspirated, 50 µL PBS (Corning) was added to wash the cells and remove FBS, the PBS was aspirated, and 40 µL Trypsin-EDTA (Corning) was added. The cells were incubated for 5–10 mins at 37 °C to allow for detachment and separation. Following incubation, 80 µL of Dulbecco's modified Eagle media (DMEM) without phenol red (Gibco) with 10% FBS was added to inactivate the trypsin. Cells were thoroughly mixed to separate and suspend individual cells. The plate(s) were then spun down at $400 \times g$ for 4 mins, and the leftover media was aspirated. Cells were resuspended in 170 µL flow buffer (PBS supplemented with 1% BSA (Thermo Fisher), 5 mM EDTA (VWR), and 0.1% sodium azide (Sigma-Aldrich) to prevent clumping). For prepping plates of cells with larger surface areas, all volumes were scaled up in proportion to the surface area, and samples were transferred to 5 mL polystyrene FACS tubes (Falcon) after

trypsinization. For standard co-transfections, 10,000–50,000 cells were collected per sample. For the poly-transfection experiment and transfections into cells harboring an existing lentiviral integration, 100,000–200,000 cells were collected per sample.

For all experiments, samples were collected on a BD LSR Fortessa equipped with a 405 nm laser with 450/50 nm filter ("Pacific Blue") for measuring TagBFP or EBFP2, 488 laser with 530/30 filter ("FITC") for measuring EYFP or mNeonGreen, 561 nm laser with 582/15 nm filter ("PE") or 610/20nm filter ('PE-Texas Red') for measuring mKate2 or mKO2, and 640 laser with 780/60 nm filter ("APC-Cy7") for measuring iRFP720. 500-2000 events/s were collected either in tubes via the collection port or in 96-well plates via the high-throughput sampler. All events were recorded using FACSDiva version 8.0.1. Compensation was not applied until processing the data (see below).

**Flow cytometry data analysis**. Analysis of flow cytometry data was performed using our MATLAB-based flow cytometry analysis pipeline (https://github.com/Weiss-Lab/MATLAB_Flow_Analysis) v0.3-beta, compatible with MATLAB 2018a+. Basic processing steps with example data are shown in Supplementary Figure 33 and follow the procedures described previously[52]. In addition, we frequently utilized our new poly-transfection technique and associated methods[45] to characterize and optimize circuits. Poly-transfection enables rapid and accurate assessment of dose–response curves for genetic components[45], such as the kinases and phosphatases in our circuits. Full schematics describing each poly-transfection experiment are shown in the SI (e.g., Supplementary Figure 2a).

Multi-dimensional binning of poly-transfection data was performed by first defining bin edges in each dimension (i.e., for the transfection markers for each poly-transfection complex), then assigning each cell to a bin where the cell's expression of these markers was less-than-or-equal-to the high bin edges and greater-than the low bin edges. Bins with three or fewer cells were ignored (values set to NaN in the MATLAB code) to avoid skewing by outliers in sparsely-populated samples (e.g., HeLa cells). Such binning is demonstrated via the colorization of cells by their bin assignment in the SI (e.g., Supplementary Figure 2b). In order to avoid the artifact of negative fold-changes, non-positive fluorescence values were discarded prior to making measurements on binned or gated populations. In the second and third experimental repeats of the miRNA-dependent signaling/classifier data in Fig. 5 and Supplementary Figures 14–16, a newly prepared Output Marker plasmid was later discovered to have an approximately eightfold lower concentration than expected due to a measurement error on the nanodrop. To account for this, the bins for the Output Marker in those samples are shifted down by 10x (so as to match the same bin boundaries as in the first repeat).

To find the optimal ratio of components in the miR-21 sensor for high cell classification accuracy, we scanned ratios between 1000:1 to 1:1000 of K:P and output plasmid:K/P, roughly halving the ratio between steps. At each combination of ratios, a trajectory was computed and all cells within 0.25 biexponential units of the trajectory based on euclidean distance were recorded. Accuracy was computed as described below, and accuracy values were compared across all ratios for each experimental repeat. From this scanning of trajectories at different ratios of components, we found that a 1:1:0.5 ratio of K:P:Output plasmid gave the highest accuracy. This optimal trajectory was used to sub-sample cells for display in Fig. 5f and Supplementary Figure 17, finding percent positive for output in Fig. 5g and calculating ROCs/AUCs in Supplementary Figure 17.

In the case of simple co-transfections and subsampled trajectories, cells were considered to be transfected if they were positive for the output/transfection marker or the output reporter. When computing summary statistics from binned data, such thresholding is unnecessary since binning already isolates the cell sub-population for measurement.

**Calculation of cell classification metrics**. Sensitivity was defined as the percent of cells positive for the output reporter in HeLa cells transfected with the T21 circuit variant. Specificity was defined as 100 minus the percent of cells positive for the output in HeLa cells with the TFF4 variant or in HEK-293 cells with the T21 variant. The former was considered the more ideal comparison for evaluating classification performance owing to the higher overall expression of the circuit in HeLa cells compared with HEKs (Supplementary Figure 14). Accuracy was computed by averaging sensitivity and specificity.

ROC curves in Supplementary Figure 17 were generated by scanning thresholds starting at $-10^8$, then 0, then 15 log-spaced steps between $10^3$ and $10^8$. The AUCs were computed individually for each experimental repeat by trapezoidal area approximation using the MATLAB function "trapz()" (https://www.mathworks.com/help/matlab/ref/trapz.html). The AUC-like curves in Supplementary Figure 18 were computed by fitting data from each experimental repeat with a bi-normal classification model in MATLAB (see below for details of the fitting algorithm used).

**Calculation of $p$ values**. $P$ values shown in Fig. 5 were computed using the MATLAB function "ttest()" (https://www.mathworks.com/help/stats/ttest.html). Samples were paired per experimental repeat and the test was two-tailed.

**Calculation of fold-changes and robustness scores**. For quantifying the effects of EnvZ variants and perturbations, we measured fold-changes by dividing the median output level of each sample by that of the equivalent sample in the absence of the EnvZ variant or perturbation. For perturbation experiments, the level of output absent perturbation is referred to as the nominal output level.

$$\text{Fold} - \Delta(\text{Input/perturbation bin}x) = \frac{\text{Output(Input/perturbation bin}x)}{\text{Output(Input/perturbation bin}1)} \quad (4a)$$

Where $\log_2$-transformed fold-changes are shown for experiments with multiple repeats, the values shown are the mean of the $\log_2$-transformed fold-changes, rather than the $\log_2$-transformation of the mean of the fold-changes. This order of operations ensures that standard deviations of the fold-changes can be computed directly on the $\log_2$-transformed scale.

We computed robustness scores from the fold-changes using the formulae below:

$$\text{Robustness(Perturbation bin}x) = 100 \cdot (1 - |1 - \text{Fold} - \Delta(\text{Perturbation bin}x)|) \quad (5a)$$

**Quantification of cell-to-cell output variance**. To measure noise, we computed the interquartile range (IQR) of the output distributions. As we chose the median to represent the middle of the distribution, the IQR is a corresponding non-parametric measurement of noise. Since gene expression noise is approximately log-distributed, we $\log_{10}$-transformed the data prior to computing the IQR. As with calculations of the medians, negative fluorescent values were discarded when computing the IQR to avoid artifacts.

**Model fitting**. Where possible, fluorescent reporters were used to estimating the concentration of a molecular species for the purpose of model fitting.

For fitting all models, we used the MATLAB function "lsqcurvefit()" (https://www.mathworks.com/help/optim/ug/lsqcurvefit.html), which minimizes the sum of the squares of the residuals between the model and the data. In general, fits were made with cells subsampled from bins, as indicated for each figure. In Supplementary Figure 18, the fits were made using the true/false positive rates for each bin. Fits were always performed individually per experimental repeat, then means and standard deviations were computed for individual fit parameters.

The goodness of fit was measured by computing the normalized root-mean-square error CV(RMSE) using the following formula:

$$\text{CV(RMSE)} = \frac{\sqrt{\frac{1}{\bar{y}}\sum_i(y(x_i) - f(x_i))^2}}{\bar{y}}$$

Where $y(x_i)$ is the value of the data at the input value $x_i$, $\bar{y}$ is the mean of $y$ for all values of $x$, and $f(x_i)$ is the function output at input value $x_i$.

Fitting functions:

Activation of transcription by OmpR-VP64:

$$y = \alpha_0 + (\alpha - \alpha_0)\frac{x^2}{K_{1/2}^2 + x^2} \quad (6)$$

The cooperativity of OmpR was assumed to be two because it forms a dimer once phosphorylated to bind DNA[16,100].

Activation of OmpR-VP64-driven expression by kinase: (see Supplementary Note 1 for more details):

$$y = \alpha_0 + (\alpha - \alpha_0)\frac{x^2}{K_{1/2}^2 + x^2} \quad (7)$$

Deactivation of OmpR-VP64 by phosphatase:

$$y = \alpha_0 + (\alpha - \alpha_0)\frac{K_{1/2}^2}{K_{1/2}^2 + x^2} \quad (8)$$

Although OmpR-VP64 has not been completely tuned over to P-OmpR-VP64, the amount of P-OmpR-VP64 is assumed to be proportional to the level of kinase because the production rate is only dependent on the kinase. In the presence of phosphatase, the decay rate becomes overwritten by the dephosphorylation reaction. Thus, these proteins can be plugged directly into the OmpR-VP64 activation function, such that the kinase is proportional to OmpR and the phosphatase is inversely so. Because of the inversion, the phosphatase function becomes a repression-form Hill function.

The bi-normal fitting function for ROC curves is included with our MATLAB flow cytometry analysis package on GitHub ("model_ROC.mat"). In short, the measurement of the fraction of cells positive for the output reporter is assumed to follow a normal distribution with $\mu_1 = 0$ and $\sigma_1 = 1$ for the negative observations (TFF4 or HEK cells in our case) and a normal distribution with unknown $\mu_2$ and $\sigma_2$ for the positive observations (T21 in HeLa cells). $\mu_2$ and $\sigma_2$ are fit such that the true positive rate for a given false positive rate approximates that of the data.

For other comparisons where we present values of $r$ or $R^2$, the former is the Pearson's correlation computed with the MATLAB function "regression()" (https://www.mathworks.com/help/deeplearning/ref/regression.html), and the latter is the coefficient of determination between predicted and actual values.

**Reporting summary**. Further information on research design is available in the Nature Research Reporting Summary linked to this article.

## Data availability
Sequences for all plasmids used in this study are provided as GenBank files in Supplementary Data. New plasmids used in this study are available from Addgene. Raw .fcs files are available from the corresponding authors upon reasonable request. Source data are provided with this paper.

## Code availability
General MATLAB code for use in .fcs file processing and analysis are available under an open-source license in our GitHub repository at https://github.com/Weiss-Lab/MATLAB_Flow_Analysis. Specific .m scripts for each experiment are available from the corresponding authors upon reasonable request.

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

## Acknowledgements

We would like to acknowledge Douglas Lauffenburger, Ahmad Khalil, and Conor McClune for helpful discussion. Thanks to Bre DiAndreth for the MATLAB code to find optimal orthogonal TCS pairs; Jeremy Gam for the MATLAB code to fit ROC curves and plasmids; both Jin Huh and Kalon Overholt for plasmids and helpful discussion; and Nika Shakiba, Melody Wu, and Margaret Zhang for help with cloning pSub0 plasmids. This work was supported by the National Science Foundation (R.W., D.D.V.: MCB-1840257).

## Author contributions

R.D.J., Y.Q., D.D.V., and R.W. designed the study; R.W., D.D.V., and M.T.L. secured funding; R.D.J., K.I., and B.W. performed the experiments; R.D.J. and B.W. analyzed the data; Y.Q. and R.D.J. developed the mathematical models; R.D.J., Y.Q., D.D.V., and R.W. wrote the manuscript; R.D.J., Y.Q., K.I., B.W., M.T.L., D.D.V., and R.W. edited and/or reviewed the manuscript.

## Competing interests

The Massachusetts Institute of Technology has filed a patent application on behalf of the inventors (R.D.J., J.H., and R.W.) of phosphorylation-based miRNA sensor design described (US Provisional Application no. 16/528,772) and a provisional application on behalf of the inventors (R.D.J., Y.Q., D.D.V., and R.W.) of the phosphorylation-based feedback controller design described. The remaining authors declare no conflict of interest.
