## [Peer Review File · Nature Communications]

Reviewers' Comments:

Reviewer #1:

Remarks to the Author:

Jones et al. present a very interesting study in which the authors ultimately construct a closed-loop control system in mammalian cells that can maintain a set-point and reject select disturbances. Namely, In this study the authors used proteins derived from bacterial two-component signaling pathways to develop synthetic phosphorylation-based and feedback-controlled devices in HEK-293 cells. The key components of this system are isolated kinase and phosphatase proteins from bifunctional histidine kinase EnvZ, which are then used to engineer a synthetic covalent modification cycle, in which the kinase and phosphatase competitively regulate phosphorylation of the cognate response regulator OmpR, enabling analog tuning of OmpR-driven gene expression. In turn, the authors show that the phosphorylation cycle can be extended by connecting phosphatase expression to small molecule and miRNA inputs in the cell, with the latter enabling cell-type specific signaling responses and accurate cell type classification. Finally, the authors design build and test a closed loop controller by co-expressing the kinase-driven output gene with the small molecule-tunable phosphatase. In general, this is an exciting and compelling body of work – with a few blemishes that detract from an otherwise beautiful study. All things considered, this work should be of significant interest to the readership of Nature Communications, and of course the synthetic biology community – after some moderate revision.

Comment 1: In Figure 1a, the cartoon implies that the phosphorylated TF activates binding and gene expression, this important detail is buried in Supplementary Figure 5. Please communicate this information in the main text and caption of Figure 1. As TFs can initiate gene expression in a number of ways this information will reduce the point of entry and understanding for readers not in the immediate community. Figure 1b, what do the colors correspond to? I am assuming that the progressive decrease in color corresponds to increasing phosphatase levels, which is a bit counter-intuitive based on the coloring. Similar comment for Figure 1c. Perhaps a heat bar/map could be useful.

Comment 2: In Figure 1a, the authors are illustrating a fundamental control system. I suggest using the term setpoint over "ideal".

Comment 3: In Results section 2.1, please clearly state the design goal, design space, workflow and justification at the outset. This is a relatively long section and I had to read this section a couple of times to appreciate the purpose of this particular segment.

Comment 4: To improve readability please reference and organize and discuss the Supplementary Figures starting from the re-assignment of Supplementary Figure 1. The authors started by first discussing Supplementary Figure 4, and I could not help but feel that I missed something.

Comment 5: In Results section 2.1, the author state "The levels of OmpR-driven gene expression induced by full-length mutants of EnvZ are shown in Figure 2a." I believe the authors are referring to Figure 2b. Also in the subsequent statement "From this initial screen, we identified two variants, EnvZm2 [T247A] and EnvZm2[AAB], the latter having an extra DHP domain fused to EnvZ[223+]37, that induced higher levels of output expression than WT EnvZ, suggesting that their phosphatase activity is reduced." While I can see the data for EnvZm2 [T247A] – orange, I am assuming the data for EnvZm2[AAB] is given in Supplementary Figure 5d? In general I found that reconciling the data, text, design space, and workflow in Section 2.1 required a significant amount of effort. Likewise, this section should be better organized and clearer given that this is the first section of the results section.

Comment 6: Supplementary Figure 5b (and other figures in the supplement presented in a similar way) what do the various colors indicate / correspond to?

Comment 7: In Results section 2.2, the author state "To evaluate the tunability of our engineered CMC, we compared the level of OmpR-VP64-driven output across combinations of kinase and phosphatase levels, with the phosphatase level regulated at the DNA and protein levels (Figure 3b-d)." I did not see a Figure 3d, please correct.

Comment 8: In Results section 2.3, the author state "In addition to ectopically-expressed factors, endogenous cellular factors can also be plugged in as inputs to the kinase (uK) and phosphatase (uK) in our engineered CMC, enabling device performance to be tuned based on factors such as the state of the cell." I am assuming this should read "the kinase (uK) and phosphatase (up)", please correct.

Comment 9: In Section 2.4 and Figure 5a, the authors present a block diagram of the feedback controller design in which a cell-free (TX/TL) processing step is indicated. However, the role of the TX/TL process is not apparent or described sufficiently in the results section.

Comment 10: Given that the authors are not genome integrating the control systems, please discuss the limits of transfection on the performance of the controllers.

Reviewer #2:

Remarks to the Author:

The authors describe an orthogonal kinase/phosphatase system derived from the bacterial two-component system that is able to drive gene expression in mammalian cells, thus laying the groundwork to reconstruct orthogonal synthetic signalling pathways in mammalian cells. The authors perform a steady-state characterisation of this system and also implement two proof-of-principle applications of the technology, one as a cell classifier using miRNA, a concept originally proposed by one of the authors (RW), and the other to implement a negative feedback controller of gene expression.

Overall the manuscript presents an original idea and provides the research community with a promising new tool for mammalian synbio.

My major observation is the lack of a dynamic characterisation of the system especially in the case of the feedback controller. After all signalling pathways have the advantage of being faster than transcription/translation based ones and thus it would be interesting to check the dynamics of this implementation. One easy way to do this could be achieved by performing an inducible perturbation by switching the pEF1a promoter of the Nanoluc and or GAL4 in Fig 6b with a CMV_TET promoter and follow fluorescent expression in single cells in timelapse microscopy. Another reason to check the dynamics in the case of the controller is that negative feedback can speed up a system's response and this capability could offset the disadvantage of the reduced dynamic range caused by the negative feedback.

It would also be interesting in the discussion to comment about the use of this system to check the hypothesis that futile phosphorylation and dephosphorylation cycles of a TF can act as an insulator against retroactivity as originally proposed by one of the authors (DDV).

Minor points:

line 192: reference should be to Figure 4c and not to 4d

line 194: same as above

line 226: a relevant reference to cell-cell variability and the role of feedback that could be added is

PMID: 24077216

REVIEWER COMMENTS

Thank you to both reviewers for your helpful feedback! Our responses are in blue text.

Reviewer #1 (Remarks to the Author):

Jones et al. present a very interesting study in which the authors ultimately construct a closed-loop control system in mammalian cells that can maintain a set-point and reject select disturbances. Namely, In this study the authors used proteins derived from bacterial two-component signaling pathways to develop synthetic phosphorylation-based and feedback-controlled devices in HEK-293 cells. The key components of this system are isolated kinase and phosphatase proteins from bifunctional histidine kinase EnvZ, which are then used to engineer a synthetic covalent modification cycle, in which the kinase and phosphatase competitively regulate phosphorylation of the cognate response regulator OmpR, enabling analog tuning of OmpR-driven gene expression. In turn, the authors show that the phosphorylation cycle can be extended by connecting phosphatase expression to small molecule and miRNA inputs in the cell, with the latter enabling cell-type specific signaling responses and accurate cell type classification. Finally, the authors design build and test a closed loop controller by co-expressing the kinase-driven output gene with the small molecule-tunable phosphatase. In general, this is an exciting and compelling body of work – with a few blemishes that detract from an otherwise beautiful study. All things considered, this work should be of significant interest to the readership of Nature Communications, and of course the synthetic biology community – after some moderate revision.

Comment 1: In Figure 1a, the cartoon implies that the phosphorylated TF activates binding and gene expression, this important detail is buried in Supplementary Figure 5. Please communicate this information in the main text and caption of Figure 1. As TFs can initiate gene expression in a number of ways this information will reduce the point of entry and understanding for readers not in the immediate community.

Thank you for pointing out this possible point of confusion – it is certainly the case that such TFs could act as either activators or repressors! We have edited the text/caption as requested to clarify.

Figure 1b, what do the colors correspond to? I am assuming that the progressive decrease in color corresponds to increasing phosphatase levels, which is a bit counter-intuitive based on the coloring. Similar comment for Figure 1c. Perhaps a heat bar/map could be useful.

The intuition of the color progression is that increasing *brightness* corresponds to increasing phosphatase activity, similar to many heatmap color scales. We have edited the caption to clarify this as we see now how different conventions for brightness/depth of color can cause confusion!

Comment 2: In Figure 1a, the authors are illustrating a fundamental control system. I suggest using the term setpoint over “ideal”.

Thank you for the suggestion – we have made the recommended change

Comment 3: In Results section 2.1, please clearly state the design goal, design space, workflow and justification at the outset. This is a relatively long section and I had to read this section a couple of times to appreciate the purpose of this particular segment.

We agree that this section needed significant clarification. To do so, we broke it in half, separating our initial efforts at mutating and modifying EnvZ from the later efforts at fixing its rotational conformation. In the first section, we added more details about the overall design goal and rationale for our approach. We have in turn broken Figure 2 into two separate figures (revised Figures 2 and 3) and brought up some elements from SI figures to more clearly illustrate the purpose and workflow of each section.

Comment 4: To improve readability please reference and organize and discuss the Supplementary Figures starting from the re-assignment of Supplementary Figure 1. The authors started by first discussing Supplementary Figure 4, and I could not help but feel that I missed something.

Thank you for pointing out this oversight! We have moved Supp Figures 1-3 to their proper position given their timing of introduction in the text.

Comment 5: In Results section 2.1, the authors state “The levels of OmpR-driven gene expression induced by full-length mutants of EnvZ are shown in Figure 2a.” I believe the authors are referring to Figure 2b.

Thank you for pointing this out – we have fixed the reference

Also in the subsequent statement “From this initial screen, we identified two variants, EnvZm2 [T247A] and EnvZm2[AAB], the latter having an extra DHP domain fused to EnvZ[223+]37, that induced higher levels of output expression than WT EnvZ, suggesting that their phosphatase activity is reduced.” While I can see the data for EnvZm2 [T247A] – orange, I am assuming the data for EnvZm2[AAB] is given in Supplementary Figure 5d? In general I found that reconciling the data, text, design space, and workflow in Section 2.1 required a significant amount of effort. Likewise, this section should be better organized and clearer given that this is the first section of the results section.

To help address this issue, as mentioned above, we have moved more results from SI figures to the main text to put the important information more squarely in front of the readers’ eyes. Now, data for induction/suppression of gene expression by all tested variants are shown in the main text.

Comment 6: Supplementary Figure 5b (and other figures in the supplement presented in a similar way) what do the various colors indicate / correspond to?

We have added a note to the figures to describe the meaning – in short, cells in each different bin are randomly assigned a color to demonstrate the sorting of cells into separate bins. In the past we have shown this with increasing color scales (1D binning) or grids of bin edges (2D), but since our results included 1D-3D binning, depending on the experiment, we decided to go with a bin illustration scheme that represented the sorting of cells into bins and the overall distribution of cells equally well across multiple dimensions of data.

Comment 7: In Results section 2.2, the author state “To evaluate the tunability of our engineered CMC, we compared the level of OmpR-VP64-driven output across combinations of kinase and phosphatase levels, with the phosphatase level regulated at the DNA and protein levels (Figure 3b-d).” I did not see a Figure 3d, please correct.

Thank you for pointing out this typo – we have fixed the reference

Comment 8: In Results section 2.3, the author state “In addition to ectopically-expressed factors, endogenous cellular factors can also be plugged in as inputs to the kinase (uK) and phosphatase (uK) in our engineered CMC, enabling device performance to be tuned based on factors such as the state of the cell.” I am assuming this should read “the kinase (uK) and phosphatase (up)”, please correct.

Thank you for pointing out this typo – we have fixed it

Comment 9: In Section 2.4 and Figure 5a, the authors present a block diagram of the feedback controller design in which a cell-free (TX/TL) processing step is indicated. However, the role of the TX/TL process is not apparent or described sufficiently in the results section.

Thank you for pointing out the potential confusion regarding cell-free TX/TL processes, which we had not realized. We have replaced “TX/TL process” with “output production”.

Comment 10: Given that the authors are not genome integrating the control systems, please discuss the limits of transfection on the performance of the controllers.

Thank you for pointing out this important point – we have added comments to the Discussion that highlight the limitations of transfection experiments, which are primarily that the dilution of DNA over time causes both the steady-state output to go to zero (preventing true steady-state measurements) and the controller performance to deteriorate as OmpR expression decreases from its peak levels.

Reviewer #2 (Remarks to the Author):

The authors describe an orthogonal kinase/phosphatase system derived from the bacterial two-component system that is able to drive gene expression in mammalian cells, thus laying the groundwork to reconstruct orthogonal synthetic signalling pathways in mammalian cells. The authors perform a steady-state characterisation of this system and also implement two proof-of-principle applications of the technology, one as a cell classifier using miRNA, a concept originally proposed by one of the authors (RW), and the other to implement a negative feedback controller of gene expression.

Overall the manuscript presents an original idea and provides the research community with a promising new tool for mammalian synbio.

My major observation is the lack of a dynamic characterisation of the system especially in the case of the feedback controller. After all signalling pathways have the advantage of being faster than transcription/translation based ones and thus it would be interesting to check the dynamics of this implementation. One easy way to do this could be achieved by performing an inducible perturbation by switching the pHEF1a promoter of the Nanoluc and or GAL4 in Fig 6b with a CMV_TET promoter and follow fluorescent expression in single cells in timelapse microscopy. Another reason to check the dynamics in the case of the controller is that negative feedback can speed up a system's response and this capability could offset the disadvantage of the reduced dynamic range caused by the negative feedback.

Thank you for bringing up dynamic characterization of the system. We first assessed system dynamics via mathematical modeling to understand whether the feedback architecture we implemented can provide faster response times. We find that it is not necessarily the case that closed loop (CL, i.e. negative feedback regulated) dynamics are faster. Specifically, according to equation (1) in the main text, the dynamics of a regulated gene can be described by

$$\begin{aligned}\frac{d}{dt}X^* &= \theta_k \frac{(X_t - X^*)K_t}{(X_t - X^*) + K_{M,k}} - \theta_p \frac{X^*P_t}{X^* + K_{M,p}}, \\ \frac{d}{dt}P_t &= \alpha\phi(X^*) - \gamma P_t,\end{aligned}$$

where P_t is the total amount of the phosphatase, whose concentration is proportional to that of the output and $\phi(X^*) = \frac{X^*/k_d}{1+X^*/k_d}$ is a Hill function describing the activation of the output/phosphatase promoter by the phosphorylated kinase. When the controller is operating in the ideal regime as a quasi-integral controller: $K_{M,p} \ll X^*$, $K_{M,k} \ll X_t$, and $X^* \ll k_d$, the dynamics of the regulated gene become:

$$\begin{aligned}\frac{d}{dt}X^* &= \theta_k K_t - \theta_p P_t, \\ \frac{d}{dt}P_t &= \alpha X^*/k_d - \gamma P_t.\end{aligned}\tag{1}$$

Since the system above is linear, we could analyze its dynamic response to a step input (K_t) by computing its eigenvalues and comparing the real part of the eigenvalues to that of an open loop system. For the CL system, the eigenvalues can be computed as:

$$S_{CL} = \frac{-\gamma \pm \sqrt{\gamma^2 - 4\theta_p\alpha/k_d}}{2} \quad (2)$$

Because θ_p is the catalytic rate constant of the de-phosphorylation reaction, it is typically several orders of magnitude larger than the protein decay rate constant γ . Assuming thus that $\gamma^2 < 4\theta_p\alpha/k_d$, we get $\max[\text{Re}(S_{CL})] = -\gamma/2$. In contrast, when the output/phosphatase is not regulated (i.e., OL system), the dynamics are described by:

$$\begin{aligned} \frac{d}{dt}X^* &= \theta_k K_t - \gamma X^*, \\ \frac{d}{dt}P_t &= \alpha X^*/k_d - \gamma P_t, \end{aligned} \quad (3)$$

whose eigenvalue is $S_{OL} = -\gamma$. Since $\max[\text{Re}(S_{CL})] > S_{OL}$, this analysis implies that the CL system does not necessarily improve system dynamics, although it enables robustness of gene expression to disturbances. This performance limitation is a well-known characteristic of integral control in general, and is the reason integral control is usually combined with proportional and derivative control in engineering [<https://press.princeton.edu/books/hardcover/9780691193984/feedback-systems>]. Recently, such a performance tradeoff has been studied for biomolecular integral controllers [<https://doi.org/10.1016/j.cels.2019.06.001>].

Experimentally, the eigenvalues computed for the ideal case above may not fully reflect the response time of the OL and CL systems. Specifically, (a) the input K_t is not a step signal but instead follows the dynamics of kinase expression, regardless of how the kinase is introduced; and (b) the controller may not be working in the ideal operation regime and therefore the assumption which simplifies the nonlinear system (1) to linear system (3) may not hold. For transient transfection experiments, plasmid dilution causes the concentration of all species to slowly converge to zero, making it relatively difficult to quantify a response time.

To analyze the impact of these factors, we ran simulations of transfections with various parameters for the OL and CL systems (**Fig R1**). The simulations show that for most of the parameters, the CL system does not respond significantly faster than the OL system to kinase input. Appreciable improvement in dynamics of the CL system may be observable when kinase plasmid concentration is low and the phosphatase plasmid concentration level is high. However, performing precise measurements in this parameter regime may be difficult, as the output level would be rather low for detection.

We followed the simulations above experimentally using transient transfection with serial flow cytometry measurements to assess the dynamics of the two systems. Consistent with our

Figure R1 | Simulations of closed loop and open loop dynamic response. Shaded areas indicate one standard deviation from the mean, where the uncertainties arise from noise in transfection efficiency. α_K and α (nM/hr) are the maximum transcription rate constants from the kinase and output promoters, respectively. They both increase linearly with the copy number of plasmids transfected at $t=0$.

analysis, the normalized dynamic outputs for various OL and CL systems (including new variants with PEST-tagged output reporters) showed no clear differences between OL and CL (Fig R2 & R3). Thus, it is likely the case that the circuit architecture and our system parameters do not provide appreciably faster dynamics in the CL configuration compared to the OL.

These experimental results do not rule out that the CL and OL dynamics differ, but at least indicate that the differences in the current design do not appear significant, at least not with transient transfections. Future studies with integrated genes and time-lapse imaging may be needed to make more precise comparisons of OL and CL responses to the kinase input and perturbations. We believe that such work necessitates a substantial new study of feedback controller dynamics across various parameter regimes, and would involve DNA circuit redesign for chromosomal integration purposes, which is beyond the scope and aims of the present manuscript. Also, future designs that incorporate proportional control may be able to provide the faster responses.

In lieu of presenting this data, we have added a few sentences to the Discussion to point out the known dynamics issues with integral feedback controllers and benefits of combining with proportional control, and to suggest studying dynamics as an exciting future area of work. We have decided not to add the rest of the above analysis to the current manuscript, as we do not yet have the experimental support needed to make strong claims about the dynamics of our feedback architecture in cells. While we can draw interesting conclusions already from the modeling results, they would be out of place in the current narrative alone.

Thank you again for the excellent suggestion to investigate our system dynamics, which has led to many interesting insights that we think will in turn lead to fruitful future research that goes well beyond our current manuscript.

It would also be interesting in the discussion to comment about the use of this system to check the hypothesis that futile phosphorylation and dephosphorylation cycles of a TF can act as an insulator against retroactivity as originally proposed by one of the authors (DDV).

Thank you for the suggestion – we have added two sentences to the Discussion as recommended.

Minor points:

line 192: reference should be to Figure 4c and not to 4d

line 194: same as above

Thank you for pointing out these typos, we have fixed them

line 226: a relevant reference to cell-cell variability and the role of feedback that could be added is PMID: 24077216

Thank you for the suggestion – we agree this is a well-suited reference and have added it

a Dynamic binning

b Output dynamics

Figure R2 (previous page) | Experimental analysis of closed loop and open loop dynamic response. HEK-293FT cells were transiently transfected with open and closed loop (OL, CL respectively) variants of the OmpR/EnvZ TCS feedback controller. In addition to the variants previously tested for perturbation rejection in the main manuscript, here we also made variants with (1) the kinase (EnvZm2) directly fused to mNeonGreen, rather than delivered on separate plasmids or 2A-linked, to more closely track kinase dynamics; and (2) +/- PEST tags on the output reporter to enable resolution of more rapid reporter dynamics. Otherwise, the experimental system was the same as in the feedback control figures in the main text; see transfection design appended to the end of the Response for more details. **(a)** To track expression over time, cells were binned into percentiles of transfection marker expression at each timepoint, such that cells with approximately the same relative dosage of kinase and output reporters could be compared over time. The percentiles were chosen by first performing standard half-log-decade binning at 48 hours, then recording the percentile of fluorescence corresponding to each bin edge. For visualization, cells in each bin are assigned a unique color. **(b)** Normalized output traces for each tested controller variant across several relevant bins containing cells with various ranges of kinase and output plasmids. All kinase dosages are shown, while only the output plasmid dosages responsive to kinase are shown. The output is normalized to the max value attained during the time course, independently for each bin for each sample. To reduce bias due to lower relative expression of CL variants compared to OL, OL samples with reduced output expression (Fluc2/3) are included and the minimum output value during the timepoint is subtracted from all other values prior to normalizing to the max. Red lines correspond to OL variants, blue to non-DDd-tagged CL variants, and green to DDd-tagged CL variants with different TMP levels administered. The results show no consistent or obviously faster time to max expression (a stand-in for steady-state given the transient nature of the experiment) for the CL vs OL variants.

Title	OmpR Controller Dynamics v1 R1				Date	2021-09-10				
Parameters										
Plate type			96-well		1 x 96-well					
Cell Types			HEK-293FT							
Cells/Well			40k							
Method			Lipofectamine 3000 (L3k)							
DNA per well (ng)					120					
Optimem (uL)					10					
# Complexes					2					
P3K Ratio (uL per ug DNA)					2					
L3K Ratio (uL per ug DNA)					2					
DNA Mix										
		Conc (ng/uL)	Size (bp)	ng/Well	Wells	Optimem	Add (uL)	P3K (uL)	L3K (uL)	Comment
Complex 1 (OmpR-VP64 + Output)				60	24	120.00		2.88	2.88	
				60	72	360.00		8.64	8.64	
Master Mix										
hEF1a:RFP720	tRJ_409	742		10.00	288	1440.00	3.88			Output Marker
hEF1a:OmpR-VP64	tRJ_924	348		10.00	288		8.27			RR Activator
Outputs										
6xOmpR-minC53:TagBFP-2A-Fluc2_1xTFF4	tRJ_1048	520	5297	25.00	24	120.00	1.15			OL
		173.33	5297	8.33	24		1.15			/3
6xOmpR-minC53:TagBFP-PEST-2A-Fluc2_1xTFF4	tRJ_925	307	5420	25.58	24		2.00			OL (PEST)
		102.23	5420	8.53	24		2.00			/3
6xOmpR-minC53:TagBFP-2A-EnvZm3t10_1xTFF4	tRJ_1049	136	4415	20.84	24		3.69			CL
6xOmpR-minC53:TagBFP-2A-DDd-EnvZm3t10_1xTFF4	tRJ_928	308	4896	23.11	72	360.00	5.41			CL (TMP-tunable)
6xOmpR-minC53:TagBFP-PEST-2A-EnvZm3t10_1xTFF4	tRJ_926	328	4541	21.43	24		1.57			CL (PEST)
6xOmpR-minC53:TagBFP-PEST-2A-DDd-EnvZm3t10_1xTFF4	tRJ_927	258	5030	23.74	72	360.00	6.64			CL (TMP-tunable, PEST)
Inert:Fluc2	tRJ_611	111.1		15.00	24		3.24			Filler DNA for outputs
		111.1		31.67	24		6.84			
		111.1		14.42	24		3.11			
		111.1		31.47	24		6.80			
		111.1		19.16	24		4.14			
		111.1		16.89	72		10.95			
		111.1		18.57	24		4.01			
		111.1		16.26	72		10.54			
Complex 2 (EnvZ Kinase)				60.00	144	720.00		17.28	17.28	
Inert:Fluc2	tRJ_611	1994.7		55.00	144		3.97			Filler DNA
hEF1a:mNeonGreen-P2A-EnvZm2	tRJ_356	93.3		5.00	144	720.00	7.72			2A-linked
hEF1a:mNeonGreen-G4S-EnvZm2	tRJ_478	331		5.00	144		2.18			Fused
TMP Levels (uM): [1e-1, 1e-2, 1e-3]										
Timepoints (hr): [4, 8, 12, 24, 48, 72]										

Figure R3 | Transfection design for measuring dynamics. Wells were simultaneously transfected, then sampled in parallel at 4, 8, 12, 24, 48, and 72 hrs post-transfection by flow cytometry (LSR-II Cytometer, Koch Institute). Output reporters were delivered at equimolar amounts to cells, with inert filler DNA used to maintain the same total mass of DNA delivered per well. Ddd-fused variants were administered TMP immediately upon transfection at the three different levels shown at the bottom, thus necessitating triple the number of wells to be transfected each. G4S refers to a GGGGS fusion linker between mNeonGreen and EnvZm2.

Reviewers' Comments:

Reviewer #2:

Remarks to the Author:

The authors satisfied all of my concerns.